# **1 Identifying ENSO Influences on Rainfall with Classification**

2 Models: Implications for Water Resource Management of Sri

## 3 Lanka

4 Thushara De Silva M.<sup>1,3</sup>, George M. Hornberger<sup>1,2,3</sup>

<sup>1</sup>Department of Civil and Environmental Engineering, Vanderbilt University, Nashville, Tennessee, USA.

<sup>2</sup>Department of Earth and Environmental Science, Vanderbilt University, Nashville, Tennessee, USA.

7 <sup>3</sup>Vanderbilt Institute for Energy and Environment, Vanderbilt University, Nashville, Tennessee, USA.

8 Correspondence to: Thushara De Silva M. (thushara.k.de.silva@vanderbilt.edu, thushara.k.de.silva@ieee.org)

9 Abstract. Seasonal to annual forecasts of precipitation patterns are very important for water infrastructure 10 management. In particular, such forecasts can be used to inform decisions about the operation of multipurpose 11 reservoir systems in the face of changing climate conditions. Success in making useful forecasts often is achieved by 12 considering climate teleconnections such as the El-Nino-Southern Oscillation (ENSO), Indian Ocean Dipole (IOD) as 13 related to sea surface temperature variations. We present a statistical analysis to explore the utility of using rainfall 14 relationships in Sri Lanka with ENSO and IOD to predict rainfall to Mahaweli and Kelani, river basins of the country. 15 Forecasting of rainfall as classes; flood, drought and normal are helpful for the water resource management decision 16 making. Results of these models give better accuracy than a prediction of absolute values. Quadratic discrimination 17 analysis (ODA) and classification tree models are used to identify the patterns of rainfall classes with respect to ENSO 18 and IOD indices. Ensemble modeling tool Random Forest is also used to predict the rainfall classes as drought and 19 not drought with higher skill. These models can be used to forecast the areal rainfall using predicted climate indices. 20 Results from these models are not very accurate; however, the patterns recognized provide useful input to water 21 resources managers as they plan for adaptation of agriculture and energy sectors in response to climate variability.

### 22 1 Introduction

The spatial and temporal uncertainty of water availability is one of the major challenges in water resource management. Understanding patterns and identifying trends in seasonal to annual precipitation are very important for

water infrastructure management. In particular, forecasts that incorporate such information can be used to inform

decisions about the operation of multipurpose reservoir systems in the face of changing climate conditions.

Success in making useful forecasts often is achieved by considering climate teleconnections such as the El-Nino-

Southern Oscillation (ENSO) as related to sea surface temperature variations and air pressure over the globe using

empirical data (Amarasekera et.al., 1997; Denise et.al., 2017; Korecha and Sorteberg, 2013; Seibert et.al., 2017). Also,

modes of variability of other tropical oceans can be related to regional precipitation (Dettinger and Diaz, 2000; Eden

et al., 2015; Maity and Kumar, 2006; Malmgren et al., 2005; Ranatunge et al., 2003; Suppiah, 1996; Roplewski and

- Halpert, 1996). For example, the effect of the Indian Ocean Dipole (IOD) is identified as independent of the ENSO
- effect (Eden et al., 2015). Pacific decadal oscillation (PDO), Atlantic multi-decadal mode oscillation (AMO), ENSO,
- and IOD teleconnections to precipitation have been found by many studies over the globe. Variations of precipitation
- in the United States are explained by ENSO, PDO and AMO (Eden et al., 2015; National Oceanic and Atmospheric

- Administration, 2017; Ward et.al., 2014), in African countries by ENSO, AMO and IOD (Reason et.al., 2006), and in
- South east Asian countries by ENSO: Indonesia (Lee, 2015; Nur'utami and Hidayat, 2016), Thailand (Singhrattna
- et.al., 2005), China (Cao et al., 2017; Ouyang et al., 2014; Qiu et.al., 2014). Australia (Bureau of Meteorology, 2012;
- Verdon and Franks, 2005), and central and south Asia (Gerlitz et al., 2016).
- The impact of ENSO and IOD on the position of the intertropical convergence zone (ITCZ) has been identified as a
- primary factor driving south Asian tropical climate variations. South Asian countries get precipitation from two
- monsoons from the movements of ITCZ in boreal summer  $(2^0 N)$  and boreal winter  $(8^0 S)$ . The South western monsoon
- (summer monsoon) is during June-August months and the North eastern monsoon (winter monsoon) is during
- December - February months (Schneider et.al, 2014). Climate teleconnections have been studied for summer
- monsoons (Singhrattna et. al., 2005; Surendran et.al., 2015) and winter monsoons (Zubair and Ropelewski, 2006), A
- negative correlation of ENSO with Indian summer monsoon has been identified (Jha et al., 2016; Surendran et al.,
- 2015).

The objective of this study is to explore the climate teleconnection to dual monsoons and inter monsoons. Water 49 resource management decisions typically are based on precipitation throughout the year and it is extremely important 50 to explore the possibility that rainfall might be related to teleconnection indices for which seasonal forecasts are 51 available. Sri Lanka is a South Asian country that gets rainfall from two monsoons and two inter-monsoons. We 52 explore ENSO and IOD climate teleconnection to Sri Lanka precipitation throughout the year. Past studies have 53 identified climate teleconnection linking precipitation to climate indices for several months and monsoon seasons, and 54 shown the importance of these for forecasting rainfall in river basins (Chandimala and Zubair, 2007; Chandrasekara 55 et al., 2003). We extend these analyses across monsoon and inter-monsoon seasons.

Although rainfall anomalies may be correlated strongly with teleconnection indices, the scatter in the data can be 57 large, making predictions from regression models have high uncertainty. However, water managers may act on 58 information about whether rainfall is expected to be abnormally low or high. Seasonal precipitation is generally 59 forecasted in broad categories. For example, the US National Weather Service forecasts seasonal precipitation as 60 above normal, below normal, and normal (National Oceanic and Atmospheric Administration, 2018). The 61 International Research Institute for Climate and Society also forecasts seasonal precipitation as above, below and near 62 normal (International Research Institute for Climate Society, 2018). We chose to follow a similar approach and 63 investigate river basin rainfall teleconnections to climate indices with classification models. If reasonably accurate 64 relationships can be developed, they will be useful for water resources management. For example, in Sri Lanka 65 decisions about allocations of water for irrigation and hydropower could be improved with estimates of when low

rainfall seasons are likely.

#### 67 2 Hydrometeorology and climatology of the study area

Sri Lanka is an island in the Indian Ocean (latitude 5° 55' N - 9° 50' N, longitudes 79° 40' E – 81° 53' E). Mean annual 69 rainfall varies from 880 mm to 5500 mm across the island. The rainfall distribution is determined by the monsoon

- system of the Indian Ocean interacting with the elevated land mass in the interior of the country. The country is divided
- into three climatic zones according to the rainfall distribution: humid zone (wet zone) (annual rainfall > 2500 mm),

intermediate zone (2500 mm < rainfall < 1750 mm) and arid zone (dry zone) (rainfall < 1750 mm) (Department of</li>
 Agriculture Sri Lanka, 2017).

Sri Lanka, a water-rich country, has 103 river basins varying from 9 km<sup>2</sup> to 10448 km<sup>2</sup>. A large fraction of the water
 resources management infrastructure of the country is associated with the Mahaweli and Kelani river basins. The

catchment areas of the Mahaweli and Kelani are 10448 km<sup>2</sup> and 2292 km<sup>2</sup> respectively. The two rivers start from the
 central highlands. Mahaweli, the longest river, travels to the ocean 331 km in the eastern direction and the Kelani 145

78 km in the western direction. Average annual discharge volume for the Mahaweli and Kelani basins are  $26368 \ 10^6 \ m^3$ 

and  $8660 \ 10^6 \text{ m}^3$  respectively (Manchanayake and Madduma Bandara, 1999). The Kelani river basin is totally inside

the humid zone whereas the Mahaweli river basin migrates through all three climate zones (Fig.1).

The temporal pattern of rainfall in Sri Lanka can be divided into four seasons as follows.

- (1) Generally low precipitation across the country from the Northeast monsoon (NEM), which gets most precipitation
- during January to February. The arid zone of the country gets significant precipitation from the NEM, whilehumid zone gets very little rainfall during this period.
- (2) The whole country gets precipitation from the first inter-monsoon (FIM) during March to April months. However,rainfall during this period is not very high across the country.
- (3) The highest precipitation for the country is from the South western monsoon (SWM) during May to September.
  However, only the humid zone gets high precipitation during this season.
- (4) The whole country gets precipitation from the second inter-monsoon (SIM) during October to December.Generally, precipitation from SIM is higher than FIM.

The time period of NEM and SIM are generally considered as December to February and October to November

respectively (Department of Meteorology Sri Lanka, 2017; Malmgren et.al, 2003; Ranatunge et al., 2003). However,

considering the bulk amount of water received from the monsoon, we consider January and February as the period of

NEM and October to December as the period of SIM.

Reflecting the rainfall seasons, the country has two agriculture seasons "Yala" (April - September) and "Maha"
(October - March). Because the arid zone gets minimal precipitation during the SWM, the agricultural systems
(165,000 ha) developed under the Mahaweli multipurpose project depend on irrigation water during the Yala season.
The country depends on stored water to drive hydropower year round. The Mahaweli and Kelani hydropower plants
of 810 MW and 335 MW capacity serve as peaking and contingency reserve power to the power system (Ceylon
Electricity Board, 2015). Management of reservoir systems is done to cater both to irrigation and hydropower
requirements.