# Peer review of "Identifying ENSO Influences on Rainfall with Classification"

_Hydrology and Earth System Sciences, 2018_

## Short Comment (SC1) · 22 Jun 2018

I believe this as a timely research for Sri Lanka as we mainly rely on the water resources for energy generation and paddy cultivation. This paper provides a reasonable application of machine learning techniques to water resources management. We all know that the identification of droughts and flood is a severe challenge to Sri Lanka. These studies will assist the water resources managers to make decisions on timely release of water and proper utilization.

---

## Short Comment (SC2) · 17 Jul 2018

Comments on "Identifying ENSO Influences on Rainfall with Classification Models: Implications for Water Resource Management of Sri Lanka" by Thushara De Silva M., George M. Hornberger:

In this paper, the authors investigates the prediction of seasonal rainfall in Sri Lanka from a machine learning perspective. The rainfall has been grouped into three classes: dry, average, and wet, in accordance with the amount of precipitation. The paper is well

organized, the structure is easy to follow, and it is a good attempt in utilizing machine learning to develop data-driven models to predict the level of precipitation over each season in the future. I suggest to address the following points before the paper can be published:

1. How to justify the selection of range of precipitation when grouping the rainfall into three classes?

2. When you choose the predicators, what is the rationality in choosing the Multivariate ENSO Index (MEI) and the Indian Ocean Dipole (IOD)? As you illustrate later, the prediction performance of the trained algorithm is not very good. May I interpret the poor performance is caused by the selection of non-informative predictors? If yes, should you consider using more informative indicators as the input of the model instead of MEI and IOD?

3. Is there any possibility to give the confidence level when making the prediction? You should also demonstrate the correlation between IOD, EMI and the quantity of interest.

4. With the prediction from the trained model, how will it facilitate the decision making in the water infrastructure management? Considering the prediction performance of the trained model, what is the risk the decision maker must carry on when making the decision? Is there any possible means to reduce the risk involved in decision making?

5. It will be better if you can highlight the contributions you have made in this paper with several bullets. This will help readers to quickly grasp the highlights in your paper.

---

## Author Comment (AC1) · 18 Jul 2018

Thank you for noticing the importance the study for Sri Lanka. Mahaweli and Kelani river basin water resources management are planned for two six months agricultural seasons, which is related to two monsoons and two inter-monsoons. Hence, identifying drought at reasonable accuracy using classification models reduces the economic losses, and both agriculture and energy sector can adapt to the climate variability. Study result motivate that research study can be continued by incorporating areal rainfall of other sub basins and long period of empirical data. In addition,it is possible study

other river basin and shorter time period.

---

## Author Comment (AC3) · 18 Jul 2018

Thank you for very valuable comments. Clarification for discussion points are given below.

1. How to justify the selection of range of precipitation when grouping the rainfall into three classes? Grouping of rainfall into three classes is based on our asumptions of water resources management. We assumed water managers would consider rainfall between -0.5xStandard Deviation (SD) and 0.5xSD as average rain and other two ends

[Figure]

as dry and wet. Selection of three rainfall classes are not motivated by statistical considerations. 2. When you choose the predicators, what is the rationality in choosing the Multivariate ENSO Index (MEI) and the Indian Ocean Dipole (IOD)? As you illustrate later, the prediction performance of the trained algorithm is not very good. May I interpret the poor performance is caused by the selection of non-informative predictors? If yes, should you consider using more informative indicators as the input of the model instead of MEI and IOD? There are several climate indices representing El-Nino-Southern Oscillation(ENSO), Indian Ocean Dipole (IOD), Pacific decadal oscillation (PDO) and Atlantic multi-decadal mode oscillation (AMO). We explore teleconnections using several climate indices representing ENSO phenomena such as MEI, NINO 4, NINO 3.4 and NINO3, and found out that MEI is a suitable index. Exploring several indices and literature survey, we selected MEI and DMI to represent ENSO and IOD for climate teleconnection of rainfall of Sri Lanka. 3. Is there any possibility to give the confidence level when making the prediction? You should also demonstrate the correlation between IOD, EMI and the quantity of interest. Correlation between IOD, DMI and seasonal rainfall demonstrate that negative correlation for three rainfall seasons (NEM, FIM and SIM) and positive correlation for SIM. Rainfall classification models identified with QDA, Classification tree and random forest also agree with this. Accuracy level of the models are shown in the Table 2, Table 3, Table 4, Table A.2-6

4. With the prediction from the trained model, how will it facilitate the decision making in the water infrastructure management? Considering the prediction performance of the trained model, what is the risk the decision maker must carry on when making the decision? Is there any possible means to reduce the risk involved in decision making? In our study, Mahaweli and Kelani river basin main water users are potable water, hydropower and agricultural systems. Water managers' decisions are informed for the water allocation for these two sectors. For example, if drought is forecasted, hydropower production can be reduced to store the water for potable water and agriculture to reduce the damages during drought. During that period, it is possible to do the maintenance of Hydropower plant machineries. Farmers can select less water intensive crops instead of high water intensive crops. Water managers are informed about the accuracy level of models, hence flexibility of emergency plans are available as the usual practice. Quantification of the risk decision maker must carry is not our scope of the study. However, updates of short term weather forecasts and emergency plans will be reduced the risk. Presently, no seasonal ahead forecast about the extremely low rainfalls is available, and this study assists to water managers for preparation of climate variability. 5. It will be better if you can highlight the contributions you have made in this paper with several bullets. This will help readers to quickly grasp the highlights in your paper. Thank you, we will consider the possibility with the manuscript format of journal.

---

## Short Comment (SC3) · 24 Jul 2018

This paper attempts to apply statistical learning models to classifying rainfall in two river basins of Sri Lanka using the relationship between rainfall in Sri Lanka and El-Nino-Southern Oscillation (ENSO), Indian Ocean Dipole (IOD). I believe the predictions in this study is helpful to the management of the water systems considering the impacts of climate change. My comments for this manuscript are as follows:

1. Line 114-116. It is mentioned that the data will be tested for normality. If not, the data

[Figure]

will be transformed. Can you provide the test results and specify the transformation function?

2. Line 117: Using one's own definition of the anomaly is okay, but I think it is necessary to justify the cutting values of the three classes of anomalies.

3. Section 2.3: Before applying the three models, can you provide the reasons why these three particular models were selected to perform the analysis? Also, I believe a bit more details about the three models are necessary in order to help more readers to understand how these models work. If possible, can you use graphs to illustrate the models?

4. Please give more details with regard to the parameter tuning in building the random forest model.

5. Can you specify the programming tools you used to perform the analysis?

6. Line 186, it would be better to increase the font size of Fig.3.

---

## Short Comment (SC4) · 28 Aug 2018

Very good and creative view to environment problem.

---

## Referee Comment (RC1) · Anonymous Referee #1 · 16 Nov 2018

General comments:

This study demonstrates application of different classification models to predict monthly rainfall using climate indices MEI and DMI. The findings of this study will be highly relevant for water managers of Sri Lanka. Below are my comments for Authors. Some of the similar comments I found that Authors have already addressed in the Discussion forum but please address all the points below.

Specific comments: 1) Line 95 says that the river basin rainfall was calculated using

[Figure]

the Thiessen polygon method. Why not divide the basin into sub-basins (using any GIS tool) based on digital elevation model and use sub-area averaged rainfall? Is this choice due to the fact that there are many reservoirs in the basin? Please clarify.

Out of 16 polygons in Mahaweli river basin and 11 in Kelani, what was the basis of selecting only 8 sub-basins?

2) Lines 104 to 109 describe how anamolies were calculated. Did you apply any of the transforms mentioned in line 108 to get normally distributed rainfall? Some plots/results can be included to clarify the rainfall anomaly classification. In Table1, the use of 0.5 appears like a random choice. Please justify.

3) Line 122 says average of MEI and DMI were used but the figures 4 and 5 show that you have used them separately. Authors should support the choice of MEI and DMI over several other climate indices which they could have used as predictors.

4) 64 years of historical data have been used, 75% of which are used for training and rest for testing model performance. If I understood properly, there is no demonstration of season-ahead forecast of rainfall and how those can be classified as dry or wet, the information useful for water managers. Authors write about forecast in Lines 240 to 246, but there is no assurance of enhancement in future skill using the three classification models used in this study.

5) Water managers will be mostly interested in extreme events. Would it be possible to obtain information about extreme dry or wet season/months from the three classification models used here?

Technical corrections:

(i) There are nomenclatures like dry and wet which are used for dividing the zones and also for classifying rainfall anomaly (see Lines 100, 160 to 177). It would be better if Authors can use different nomenclature.

(ii) In Figure 4, caption of part (d) is missing.

---

## Referee Comment (RC2) · Anonymous Referee #2 · 29 Nov 2018

See attached pdf

[Figure]

This paper examines the use of climate indices to predict a high or low rainfall period. Classifiation tools are used for this.

The results that are obtained are not very impressive, but the authors argue that, for the local farmers and water managers, this will still be of value, which is a fair point. So while the scientific interest of this paper is limited, it has some clear practical value.

The paper in its current form suffers from: (i) an insufficiently detailed presentation of the methodology which would not enable a reader, even in principle, to understand how the methods work unless the reader had prior knowledge of them; (ii) a strange organisation of the material so that the presentation of the study area is given under a 'methods' section for instance. This may be because the authors seem to be wedded to organising the paper according to some standard headings: methods, results, discussion, etc. But this is not always helpful and here, as with the obligatory conclusion section which is just a repetition of material just above that section, I would urge the authors to feel free to adapt the structure to their needs.

Some detailed comments follow.

| Page | Line | Comment |
|---|---|---|
| 1 | 21 | something missing in this sentence, perhaps 'to' before 'climate variability' (without 'the') |
| 2 | 56-61 | It is not clear here whether you are making a methodological point here. It seems that you are identifying two reasons for your methodological approach: (i) the weakness of the linear regression approach when the scatter is large and (ii) the nature of the forecasts available to water managers, which may just be of some broad category of rainfall rather than actual quantities. Based upon these two reasons, you are advocating a method based upon classification models. If that is the case, please spell this out as these are key issues for understanding your chosen approach. |
| 2-5 | | Secontion 2 until the middle of page 5 (the start of subsection 2.3) is not about methods. Please choose a more appropriate title for the section, such as 'Hydrometeorology and climatology of the study area'. Subsection 2.3 can then become a section 3 entitled 'Methods'. |
| 5 | 117-8 | I am not sure why you mention a minimum and a maximum in the table. Given that we have no idea what these might be, I suggest taking out any reference to them (so the first class is just defined for standardised anomalies below -0.5C, and similarly for the third class with standardised anomalies large then 0.5C) |
| 5 | 122 | Capital letters are required here: 'Atmospheric' and 'Administration' as well as a comma after the latter word |
| 6 | 138-45 | To make this presentation of the QDA clearer to someone who has, for instance, some idea of Bayesian statistics, but does not know this method, I suggest adding the the quadratic discriminant function is therefore proportional to the logarithm of the a posteriori density function of class $k$ conditional upon the value of the observed predictor $x$: this logarithm is the product of the prior probability of $x$ and the density |

**Fig. 1.**

---

## Author Comment (AC4) · 3 Dec 2018

**Reviewer comments are shown in blue color fonts,** and authors' responses are given black color fonts.

General comments:
This study demonstrates application of different classification models to predict monthly rainfall using climate indices MEI and DMI. The findings of this study will be highly relevant for water managers of Sri Lanka. Below are my comments for Authors. Some of the similar comments I found that Authors have already addressed in the Discussion forum but please address all the points below.

**Response:** Thank you for identifying the importance of the study. Shortcomings of the paper identified by the reviewers will be rectified as suggested, for the next submission. Response for the detail comments are given below.

Specific comments:

1) (a) Line 95 says that the river basin rainfall was calculated using the Thiessen polygon method. Why not divide the basin into sub-basins (using any GIS tool) based on digital elevation model and use sub-area averaged rainfall? Is this choice due to the fact that there are many reservoirs in the basin? Please clarify.

**Response:** The division of river basin into sub basins was limited by the information availability from the water resources agencies of Sri Lanka. This study does not aim to explore rainfall differences across sub-basins, so we did not need to define sub basins using DEMs.

(b) Out of 16 polygons in Mahaweli river basin and 11 in Kelani, what was the basis of selecting only 8 sub-basins?

**Response:** The two largest reservoir cascades of Sri Lanka are associated with Mahaweli and Kelani river basins. When selecting the 8 sub-basins from two river basins, we looked at the most important sub-basins for the reservoir catchment as well as water use. Kelani river basin's sub-basins: Laxapana, Norwood, Norton Bridge are main catchments for the two largest reservoirs (Moussakele and Norton Bridge) at the upstream of the reservoir cascade. For the Mahaweli, Morape represents the Kotmale reservoir catchment, Peradeniya represents the Victoria reservoir, Randenigala represents the Randenigala reservoir, Bowatenna represents the Bowatenna reservoir and Manampitiya represents the dry zone agricultural tanks.

We can add one or two sentence to the manuscript clarifying the basis for selecting 8 sub-basins.

2) (a) Lines 104 to 109 describe how anomalies were calculated. Did you apply any of the transforms mentioned in line 108 to get normally distributed rainfall? Some plots/results can be included to clarify the rainfall anomaly classification.

**Response:** Shapiro-Wilk's method is used to identify the normality of rainfall anomaly distribution. For example, Manampitiya NEM normality test results are given below.

Data 1: original data
W = 0.96675, p-value = 0.08185

Data 2: data transformed by square root
W = 0.98772, p-value = 0.7772
Data 3: data transformed by log
W = 0.91577, p-value = 0.0003325
Further, from data plots (Figure S 1) and the S-W stastic, we conclude that the square root transfo rmed data is closer to being normally distributed than the other forms.

[Figure]

Figure S 1: Manampitiya NEM standardized data (a) original form qqplot (b) square root form qqplot (c) original form density plot (d) square root form density plot

(b) In Table1, the use of 0.5 appears like a random choice. Please justify.

**Response:** The choice of 0.5 (or any cutoff) is a choice. Using 0.5 as a threshold for a normal distribution defines portions of the data that are fairly evenly distributed into three categories – about 31%, 38%, and 31% for a normal distribution. We deemd this a reasonable choice for our analysis.

3) (a) Line 122 says average of MEI and DMI were used but the figures 4 and 5 show that you have used them separately.

**Response:** MEI and DMI values are originally in a monthly time step. Since we analyzed the data in rainfall seasons, MEI and DMI values over the season are averaged. For example for the NEM season, the MEI value is the average of January and February monthly values. For SWM season, DMI is the average of May, June, July and September values.

(b) Authors should support the choice of MEI and DMI over several other climate indices which they could have used as predictors.

**Response:**

[Figure]

Figure S 2: Correlation between Norwood rainfall anomalies with multiple climate indices

Table S 1: Correlation analysis of rainfall anomalies and climate indices

| Rainfall | Morape | | | | | Peradeniya | | | | |
|---|---|---|---|---|---|---|---|---|---|---|
| Month | MEI | NINO34 | NINO3 | NINO4 | *DMI* | MEI | NINO34 | NINO3 | NINO4 | *DMI* |
| NEM | -0.35 | -0.35 | -0.34 | -0.38 | *-0.09* | -0.38 | -0.40 | -0.39 | -0.42 | *-0.11* |
| FIM | -0.28 | -0.19 | -0.28 | -0.07 | *-0.11* | -0.27 | -0.18 | -0.30 | -0.06 | *-0.06* |
| SWM | -0.35 | -0.24 | -0.23 | -0.26 | *-0.29* | -0.35 | -0.26 | -0.25 | -0.27 | *-0.31* |
| SIM | 0.21 | 0.23 | 0.27 | 0.19 | *0.12* | 0.17 | 0.19 | 0.21 | 0.15 | *0.09* |
| Rainfall | Laxapana | | | | | Norwood | | | | |
| Month | MEI | NINO34 | NINO3 | NINO4 | *DMI* | MEI | NINO34 | NINO3 | NINO4 | *DMI* |
| NEM | -0.27 | -0.26 | -0.28 | -0.27 | *-0.01* | -0.28 | -0.26 | -0.29 | -0.27 | *-0.04* |
| FIM | -0.28 | -0.16 | -0.27 | -0.03 | *-0.07* | -0.27 | -0.18 | -0.26 | -0.03 | *-0.13* |
| SWM | -0.3 | -0.23 | -0.21 | -0.25 | *-0.31* | -0.21 | -0.12 | -0.15 | -0.16 | *-0.24* |
| SIM | 0.1 | 0.10 | 0.14 | 0.06 | *0.08* | 0.29 | 0.31 | 0.32 | 0.27 | *0.28* |
| Rainfall | Randenigala | | | | | Bowatenna | | | | |

| Month | MEI | NINO34 | NINO3 | NINO4 | *DMI* | MEI | NINO34 | NINO3 | NINO4 | *DMI* |
|---|---|---|---|---|---|---|---|---|---|---|
| NEM | -0.30 | -0.31 | -0.29 | -0.34 | *-0.11* | -0.35 | -0.36 | -0.35 | -0.38 | *-0.2* |
| FIM | -0.29 | -0.23 | -0.33 | -0.10 | *-0.04* | -0.23 | -0.17 | -0.25 | -0.09 | *-0.02* |
| SWM | -0.17 | -0.12 | -0.09 | -0.18 | *-0.24* | -0.18 | -0.09 | -0.05 | -0.11 | *-0.12* |
| SIM | 0.37 | 0.38 | 0.41 | 0.36 | *0.35* | 0.35 | 0.41 | 0.40 | 0.40 | *0.36* |
| **Rainfall** | **Norton Bridge** | | | | | **Manampitiya** | | | | |
| Month | MEI | NINO34 | NINO3 | NINO4 | *DMI* | MEI | NINO34 | NINO3 | NINO4 | *DMI* |
| NEM | -0.32 | -0.30 | -0.33 | -0.33 | *-0.01* | -0.26 | -0.28 | -0.26 | -0.28 | *-0.16* |
| FIM | -0.18 | -0.12 | -0.21 | -0.01 | *-0.08* | -0.2 | -0.17 | -0.31 | -0.06 | *-0.14* |
| SWM | -0.31 | -0.22 | -0.21 | -0.22 | *-0.37* | -0.07 | 0.08 | 0.08 | -0.01 | *-0.03* |
| SIM | 0.02 | -0.02 | 0.03 | -0.04 | *-0.15* | 0.45 | 0.46 | 0.44 | 0.46 | *0.51* |

We examined the correlation between rainfall anomalies and multiple climate indices to choose the two climate indices MEI and DMI (Figure S *2*, Table S 1). ENSO phenomenon is represented by MEI, NINO34, NINO3, NINO4 indices, and Indian Ocean dipole phenomenon is represented by DMI index. Correlation analysis indicates that MEI and sub-basins' rainfall anomalies demonstrates higher correlation for all the rainfall seasons compared to the NINO34, NINO3 and NINO4. DMI represents the intensity of IOD, the gradient of the sea surface temperature. MEI is estimated using several climate factors such as sea-level pressure, zonal and meridional components of the surface wind, sea surface temperature, surface air temperature, and total cloudiness fraction of the sky (National Oceanic and atmospheric administration 2017). Therefore, considering high correlation values between MEI and rainfall anomalies as well as MEI based on several climate parameters in addition to the sea surface temperature, we selected MEI as the indicator for ENSO with DMI as the indicator for IOD.

> 4) 64 years of historical data have been used, 75% of which are used for training and rest for testing model performance. If I understood properly, there is no demonstration of season-ahead forecast of rainfall and how those can be classified as dry or wet, the information useful for water managers. Authors write about forecast in Lines 240 to 246, but there is no assurance of enhancement in future skill using the three classification models used in this study.

**Response:** As indicated in the Lines 253-258; ENSO forecasts are available from the International Research Institute for Climate and Society (International Research Institute, 2017b) and IOD forecasts are available in the Bureau of Meteorology (BOM), Australian Government (Bureau of Meteorology, 2017). We do not know of the availability of archives of MEI and DMI past forecasts that could be used to evaluate the skill of the season-ahead forecasts of rainfall. Thus we could not evaluate the degree to which our analysis would improve overall skill.

5) Water managers will be mostly interested in extreme events. Would it be possible to obtain information about extreme dry or wet season/months from the three classification models used here?

**Response:** Because our analysis is for seasonal precipitation, we think it is most appropriate for considering dry conditions. Extreme wet conditions are most important for flooding in this region and such analyses would best be done on a much shorter time scale. For very low conditions, the QDA method produces results that are promising (Table S2).

Table S 2: Extreme dry (very low rainfall) and wet (very high rainfall) events identifying skill

| Class | Range | NortonBridge SWM | | Manampitiya NEM | |
|---|---|---|---|---|---|
| | | tree | QDA | tree | QDA |
| Very low | $X_{S\_ANM} < -1.0$ | 10/11 | 10/11 | 6/11 | 11/11 |
| low | $-1.0 <= X_{S\_ANM} < -0.5$ | 9/11 | 6/11 | 5/11 | 9/10 |
| normal | $-0.5 <= X_{S\_ANM} < 0.5$ | 8/22 | 9/22 | 9/25 | 11/25 |
| high | $0.5 <= X_{S\_ANM} <= 1.0$ | 5/11 | 5/11 | 1/5 | 0/5 |
| Very high | $1.0 <= X_{S\_ANM}$ | 6/11 | 6/11 | 7/11 | 1/11 |

Technical corrections:

(i) There are nomenclatures like dry and wet which are used for dividing the zones and also for classifying rainfall anomaly (see Lines 100, 160 to 177). It would be better if Authors can use different nomenclature.

**Response:** We will use three rainfall classes as high, normal and low instead of dry, average and wet.

(ii) (ii) In Figure 4, caption of part (d) is missing.

**Response:** Caption will corrected as including (d) (b) Manampitiya NEM rainfall classification by QDA

**Reference:**

Bureau of Meteorology. (2017). Indian Ocean, POAMA monthly mean IOD forecast. Retrieved March 30, 2017, from http://www.bom.gov.au/climate/enso/#tabs=Indian-Ocean

International research institute. (2017). IRI ENSO forecast. Retrieved January 1, 2017, from http://iri.columbia.edu/our-expertise/climate/forecasts/enso/current

National oceanic and atmospheric administration. (2017). Cold & Warm Episodes by Season. Retrieved March 30, 2017, from http://www.cpc.ncep.noaa.gov/products/analysis_monitoring/ensostuff/ensoyears.shtml

---

## Author Comment (AC5) · 3 Dec 2018

**Reviewer comments are shown in blue color fonts,** and authors' responses are given black color fonts.

This paper examines the use of climate indices to predict a high or low rainfall period. Classification tools are used for this. The results that are obtained are not very impressive, but the authors argue that, for the local farmers and water managers, this will still be of value, which is a fair point. So while the scientific interest of this paper is limited, it has some clear practical value. The paper in its current form suffers from: (i) an insufficiently detailed presentation of the methodology which would not enable a reader, even in principle, to understand how the methods work unless the reader had prior knowledge of them; (ii) a strange organisation of the material so that the presentation of the study area is given under a 'methods' section for instance. This may be because the authors seem to be wedded to organising the paper according to some standard headings: methods, results, discussion, etc. But this is not always helpful and here, as with the obligatory conclusion section which is just a repetition of material just above that section, I would urge the authors to feel free to adapt the structure to their needs.

**Response:** The objective of this study is to explore the climate teleconnection to dual monsoon and inter monsoon and provide information for water resources managers and water users (farmers and hydropower producers) decisions considering forecasted climate indices. Specifically for our case study of Sri Lanka, presently there is no seasonal precipitation forecast and this information will be informed water resources planning and water users' climate adaptation decisions.

Shortcomings of the paper identified by the reviewers will be rectified as suggested, for the next submission. Response for the detail comments are given below.

Detailed Comments:

1. something missing in this sentence, perhaps 'to' before 'climate variability' (without 'the') (page 1, line 21)

**Response:** This sentence will be corrected as below:
"Results from these models are not very accurate; however, the patterns recognized are useful input to the water resources management and to the adaptation climate variability of agriculture and energy sectors."

2. It is not clear here whether you are making a methodological point here. It seems that you are identifying two reasons for your methodological approach: (i) the weakness of the linear regression approach when the scatter is large and (ii) the nature of the forecasts available to water managers, which may just be of some broad category of rainfall rather than actual quantities. Based upon these two reasons, you are advocating a method based upon classification models. If that is the case, please spell this out as these are key issues for understanding your chosen approach. (page 2, line 56-61)

**Response:** Seasonal precipitation is generally forecasted in broad categories. For example, the US National Weather Service forecasts seasonal precipitation as above normal, below normal, and normal (National Oceanic and Atmospheric Administration, 2018). The International Research Institute for Climate and Society also forecasts seasonal precipitation as above, below and near

normal (International Research Institute for Climate Society, 2018). We chose to follow a similar approach and present seasonal precipitation prediction in three classes based on ENSO and IOD.

3. Secontion 2 until the middle of page 5 (the start of subsection 2.3) is not about methods. Please choose a more appropriate title for the section, such as 'Hydrometeorology and climatology of the study area'. Subsection 2.3 can then become a section 3 entitled 'Methods'. (page 2-5)

**Response:** We will reorganize the structure of the paper as suggested, in the final submission.

4. I am not sure why you mention a minimum and a maximum in the table. Given that we have no idea what these might be, I suggest taking out any reference to them (so the first class is just defined for standardised anomalies below -0.5C, and similarly for the third class with standardised anomalies large then 0.5C) (page 5, line 117-118)

**Response:** The Table was revised as given below. Originally, the terms maximum and minimum referred to the largest observed positive anomaly value and largest negative anomaly value respectively but the reviewer points out that these terms are misleading so they were dropped.

Table 1

| Class | Range |
|-------|-------|
| below | $X_{S\_ANM} < -0.5$ |
| normal | $-0.5 <= X_{S\_ANM} < 0.5$ |
| above | $0.5 <= X_{S\_ANM}$ |

5. Capital letters are required here: 'Atmospheric' and 'Administration' as well as a comma after the latter word. (page 5, line 122)

**Response:** Word will be corrected as "National Oceanic and Atmospheric Administration".

6. To make this presentation of the QDA clearer to someone who has, for instance, some idea of Bayesian statistics, but does not know this method, I suggest adding the quadratic discriminant function is therefore proportional to the logarithm of the a posteriori density function of class $k$ conditional upon the value of the observed predictor $x$: this logarithm is the product of the prior probability of $x$ and the density. (page 6, line 138-145)

**Response:** A brief outline of the Quadtratic Districriminant Analysis method is given below. We can include this information in the methods part of the paper or possibly in an Appendix to the paper. More information about the methods can be obtained from the references as indicated in the paper.

"The mathematical formulation of QDA can be derived from Bayes theorem assuming that observations from each class are drawn from a Gaussian distribution ((James, Witten, Hastie, & Tibshirani, 2013; Löwe, Madsen, & McSharry, 2016).
The prior probability $\pi_k$ represents the randomly chosen observation coming from kth class with density function $f_k(x)$. Bayes theorem states that

$$Pr(Y = k|X = x) = -\frac{\pi_k f_k(x)}{\sum_{l=1}^{K} \pi_l f_l(x)} \qquad \text{Eq.(1)}$$

In Eq (1), the posterior probability $Pr(Y = k|X = x)$ indicates that observation $X = x$ belongs to the kth class. For p predictors, the multivariate Gaussian distribution density function is defined for every class k (Eq.(2)).

$$f_k(x) = -\frac{1}{(2\pi)^{p/2}|\Sigma_k|^{1/2}} \, exp\left(-\frac{1}{2}(x - \mu_k)^T \Sigma_k^{-1}(x - \mu_k)\right) \qquad \text{Eq.(2)}$$

In Eq.(2), $\Sigma_k$ is the covariance matrix and $\mu_x$ is the mean vector. The covariance matrix ($\Sigma_k$) and mean ($\mu_x$) for each class are estimated from the training data set (Eq.(3), Eq.(4)).

$$\mu_k = -\frac{1}{N_k} \sum_{i:y_i=k} x_i \qquad \text{Eq.(3)}$$

$$\text{Eq.(4)}$$
$$\Sigma_k = -\frac{1}{(N_k - 1)} \sum_{i:y_i=k} (x_i - \mu_k)^T (x_i - \mu_k)$$

Substituting a Gaussian density function for the k[th] class (Eq.(2) into Bayes theorem and taking the log values, the quadratic discriminant function is derived (Eq.(5)). Prior probabilities for class k ($\pi_k$) is calculated by the frequency of data points of class k in the training data (Eq.(6)). For a total number of $N$ points in the training observations, $N_k$ is the number of observations belong to kth class.

$$\delta_k(x) = -\frac{1}{2}(x - \mu_x)^T \Sigma_k^{-1}(x - \mu_x) + log\,\pi_k \qquad \text{Eq.(5)}$$

$$\pi_k = -\frac{N_k}{N} \qquad \text{Eq.(6)}$$

Covariance, mean and prior probability values are inserted into the discriminant function ($\delta_k(x)$) together with the state variables (Eq.(5)). The corresponding class is selected according to the largest value of the function. The number of parameters to be estimated for the QDA model for k classes and p predictors is $k.p.(p + 1) / 2$. For this study, the QDA model output is the probability that an observation of a climate category will fall into each of the rainfall classes."

**References:**
International Research Institute for Climate Society. (2018). IRI Seasonal Precipitation Forecast. Retrieved February 12, 2018, from http://iridl.ldeo.columbia.edu/maproom/Global/Forecasts/NMME_Seasonal_Forecasts/Precipitation_ELR.html
James, G., Witten, D., Hastie, T., & Tibshirani, R. (2013). *Springer Texts in Statistics An Introduction to Statistical Learning - with Applications in R*. https://doi.org/10.1007/978-1-

4614-7138-7

Löwe, R., Madsen, H., & McSharry, P. (2016). Objective classification of rainfall in northern Europe for online operation of urban water systems based on clustering techniques. *Water (Switzerland)*, *8*(3). https://doi.org/10.3390/w8030087

National Oceanic and Atmospheric Administration. (2018). Three Months Outlook, Official Forecast, Climate Prediction Center, National Weather Services. Retrieved February 12, 2018, from http://www.cpc.ncep.noaa.gov/products/predictions/long_range/seasonal.php?lead=6

---

## Author Comment (AC6) · 4 Dec 2018

**Reviewer comments are shown in blue color fonts, and authors' responses are given black color fonts.**

General comments:
This paper attempts to apply statistical learning models to classifying rainfall in two river basins of Sri Lanka using the relationship between rainfall in Sri Lanka and El-Nino Southern Oscillation (ENSO), Indian Ocean Dipole (IOD). I believe the predictions in this study is helpful to the management of the water systems considering the impacts of climate change. My comments for this manuscript are as follows:

**Response:** Thank you for identifying the importance of the study. Shortcomings of the paper identified by the reviewers will be rectified as suggested, for the next submission. Response for the detail comments are given below.

> 1. Line 114-116. It is mentioned that the data will be tested for normality. If not, the data will be transformed. Can you provide the test results and specify the transformation function?

**Response:** Shapiro-Wilk's method is used to identify the normality of rainfall anomaly distribution. When the original data does not normally distribute, we carried out several transformation methods, and find the best transformation for particular rainfall data set. For example, Peradeniya NEM normality test results are given below.

Data 1: original data
W = 0.92157, p-value = 0.0005787
Data 2: data transformed by log
W = 0.92625, p-value = 0.000916
Data 3: data transformed by Square
W = 0.70236, p-value = 4.266e-10
Data 4: data transformed by Square root
W = 0.98177, p-value = 0.4632

[Figure]

Figure S 1: Peradeniya NEM data qqplots (a) original form (b) log transformation (c) square transformation (d) square root transformation

Further, from data plots (Figure S 1) and the S-W statistic, we conclude that the square root transformed data is closer to being normally distributed than the other forms. In this particular example of Peradeniya NEM season, we used square root transformed anomalies. However, for other data sets, it can be a different transformation function.

2. Line 117: Using one's own definition of the anomaly is okay, but I think it is necessary to justify the cutting values of the three classes of anomalies.

**Response:** Definition of precipitation anomaly is a choice. Extreme seasonal precipitation were defined statistically in different manner in various studies using statistical thresholds (Easterling et al., 2000; Jentsch, Kreyling, & Beierkuhnlein, 2007; Knapp et al., 2015; Smith, 2011). Using 0.5 as a threshold for a normal distribution defines portions of the data that are fairly evenly distributed into three categories – about 31%, 38%, and 31% for a normal distribution (Figure S 2). We deemed this a reasonable choice for our analysis.

[Figure]

Figure S 2: (a) Norton Bridge SWM rainfall anomaly distribution (b) Manampitiya NEM rainfall anomaly distribution

> 3. a) Section 2.3: Before applying the three models, can you provide the reasons why these three particular models were selected to perform the analysis?

**Response:** We have explored several statistical techniques to identify the relationship between rainfall anomalies and ENSO and IOD climate indices. We have used linear discriminant analysis, quadratic discriminant analysis, regression and classification tree models, Naïve Bayes, KNN and Random Forest methods. Study results indicate that quadratic discriminant analysis, classification tree models and Random Forest models are given higher accuracy than the other methods for this application.

> b) Also, I believe a bit more details about the three models are necessary in order to help more readers to understand how these models work. If possible, can you use graphs to illustrate the models?

**Response:** A brief outline of the Quadtratic Districriminant Analysis method, Classification tree and Random Forest model are given below. More information about the methods can be obtained from the references as indicated in the paper.

**2.3.1 Quadtratic Districriminant Analysis (QDA)**

The mathematical formulation of QDA can be derived from Bayes theorem assuming that observations from each class are drawn from a Gaussian distribution ((James, Witten, Hastie, & Tibshirani, 2013; Löwe, Madsen, & McSharry, 2016).

The prior probability $\pi_k$ represents the randomly chosen observation coming from kth class with density function $f_k(x)$. Bayes theorem states that

$$Pr(Y = k | X = x) = -\frac{\pi_k f_k(x)}{\sum_{l=1}^{K} \pi_l f_l(x)} \qquad \text{Eq.(1)}$$

In Eq (1), the posterior probability $Pr(Y = k|X = x)$ indicates that observation $X = x$ belongs to the kth class. For p predictors, the multivariate Gaussian distribution density function is defined for every class k (Eq.(2)).

$$f_k(x) = -\frac{1}{(2\pi)^{p/2}|\Sigma_k|^{1/2}} \, exp\left(-\frac{1}{2}(x - \mu_k)^T \Sigma_k^{-1}(x - \mu_k)\right) \qquad \text{Eq.(2)}$$

In Eq.(2), $\Sigma_k$ is the covariance matrix and $\mu_x$ is the mean vector. The covariance matrix $(\Sigma_k)$ and mean $(\mu_x)$ for each class are estimated from the training data set (Eq.(3), Eq.(4)).

$$\mu_k = -\frac{1}{N_k} \sum_{i:y_i=k} x_i \qquad \text{Eq.(3)}$$

$$\text{Eq.(4)}$$
$$\Sigma_k = -\frac{1}{(N_k - 1)} \sum_{i:y_i=k} (x_i - \mu_k)^T(x_i - \mu_k)$$

Substituting a Gaussian density function for the $k^{th}$ class (Eq.(2) into Bayes theorem and taking the log values, the quadratic discriminant function is derived (Eq.(5)). Prior probabilities for class k $(\pi_k)$ is calculated by the frequency of data points of class k in the training data (Eq.(6)). For a total number of $N$ points in the training observations, $N_k$ is the number of observations belong to kth class.

$$\delta_k(x) = -\frac{1}{2}(x - \mu_x)^T \Sigma_k^{-1}(x - \mu_x) + log\,\pi_k \qquad \text{Eq.(5)}$$

$$\pi_k = -\frac{N_k}{N} \qquad \text{Eq.(6)}$$

Covariance, mean and prior probability values are inserted into the discriminant function $(\delta_k(x))$ together with the state variables (Eq.(5)). The corresponding class is selected according to the largest value of the function. The number of parameters to be estimated for the QDA model for k classes and p predictors is $k.p.(p + 1) / 2$. For this study, the QDA model output is the probability that an observation of a climate category will fall into each of the rainfall classes."

**2.3.2 Classification Tree model**

For the classification tree model the predictor space is divided into non-overlapping regions $(R_1..R_j)$. A classification tree predicts each observation as belonging to the most commonly occurring class of the training data regions (James et.al., 2013). Recursive binary splitting is used to grow the classification tree.
Classification error rate, Gini index and cross-entropy are typically used to evaluate the quality of particular split (James et.al., 2013), and in our study we used the first two indices. Classification error rate $(E)$ gives fraction of observation that do not belong to the most commonly occurring class of the training data regions (Eq.(7)). However, for the tree-growing, the Gini index $(G)$ is considered as the criterion for splitting into regions (Eq.(8))

$$E = 1 - max_k(\hat{p}_{mk}) \hspace{3cm} \text{Eq.(7)}$$

$$G = \sum_{k=1}^{K} \hat{p}_{mk}(1 - \hat{p}_{mk}) \hspace{2.5cm} \text{Eq.(8)}$$

In Eq.(8) and Eq.(8) , $\hat{p}_{mk}$ represents the fraction of observations in the m[th] class that belong to the k[th] class. The Gini index is considered as a measure of node purity of the tree model, since small values of the index indicate that node has a higher number of observations from a single class.

The complexity of the trees are adjusted using a pruning process to produce more interpretable results. Complex trees reduces training error by overfitting the training data. Simple trees can be interpreted well, however, selecting a model which can find the pattern of data is important. In order to achieve the low classification error (training error + testing error), pruning technique is used. First, grow the very large tree, and sub tree is obtained by removing the weak links of the tree. Using tuning parameter to examine the trade-off between complexity of tree and the training error, and defining minimum samples for a node, maximum depth of the tree, and maximum number of terminal nodes are some of the pruning methods (Analytical Vidhya Team, 2016). For this study, we defined the maximum number of nodes to obtain the simple tree (pruned tree).

Tree models give the probability that an observation falls into each of the three rainfall classes. The predicted class is assigned based on the highest probability. Tree models handle ties of probability values by randomly assigning the class.

**2.3.3 Random Forest**

A random forest is an ensemble learning method used for classification and regression problems. The method is based on a multitude of decision trees based on training data with the final model as the mean of the ensemble (Breiman, 2001). Individual trees are built on a random sample of the training data with several predictors from the total number of predictors. Individual trees are built from the bootstrapped training data set.

There are some features, which can be tuned to make the better performed random forest model. Maximum number of predictors from the total predictors for individual trees, maximum number of trees, maximum node size of the trees and minimum sample leaf size are some of these features (Analytical Vidhya Team, 2015). In our study, we use the maximum number of trees as the main tuning parameters.

In a random forest model the importance of the variable is measured as the decrease in node impurity from the splits over the variable. This value is calculated by averaging the Gini index over the multitude of trees with a larger value indicating high importance of the predictor (James et.al., 2013).

4. Please give more details with regard to the parameter tuning in building the random forest model.

**Response:** We carried out the study for 100 to 5000 number of trees to build the random forest model. Highest accuracy is given by 500 numbers of trees.

5. Can you specify the programming tools you used to perform the analysis?

**Response:** We used R programming language to carry out the classification. R packages: caret, tree, randomForest, fitdistriplus, devtools and quantreg are used for the studies.

6. Line 186, it would be better to increase the font size of Fig.3.

**Response:** I will improve the Fig.3 in the final submission as given below.

[Figure]

---

## Author Comment (AC7) · 4 Dec 2018

**2.3 Statistical Analyses**

Seasonal values of MEI and DMI were used as the predictors to classify seasons into the three rainfall classes. The total data set is divided into 75 % for training the model and 25 % for testing model performance. Quadratic discriminant analysis (QDA) and classification trees were selected for the analyses. A random forest model also was applied to investigate the reliability of a cross-validated statistical forecast tool based on an advance estimate of MEI and DMI. We used R programming language to carry out the statistical analyses. R packages: caret, tree, randomForest, fitdistriplus, devtools and quantreg are used for the studies.

**2.3.1 Quadtratic Districriminant Analysis (QDA)**

The mathematical formulation of QDA can be derived from Bayes theorem assuming that observations from each class are drawn from a Gaussian distribution ((James, Witten, Hastie, & Tibshirani, 2013; Löwe, Madsen, & McSharry, 2016).
The prior probability $\pi_k$ represents the randomly chosen observation coming from kth class with density function $f_k(x)$. Bayes theorem states that

$$Pr(Y = k|X = x) = -\frac{\pi_k f_k(x)}{\sum_{l=1}^{K} \pi_l f_l(x)} \qquad \text{Eq.(1)}$$

In Eq (1), the posterior probability $Pr(Y = k|X = x)$ indicates that observation $X = x$ belongs to the kth class. For p predictors, the multivariate Gaussian distribution density function is defined for every class k (Eq.(2)).

$$f_k(x) = -\frac{1}{(2\pi)^{p/2}|\Sigma_k|^{1/2}} \, exp\left(-\frac{1}{2}(x - \mu_k)^T \Sigma_k^{-1}(x - \mu_k)\right) \qquad \text{Eq.(2)}$$

In Eq.(2), $\Sigma_k$ is the covariance matrix and $\mu_x$ is the mean vector. The covariance matrix ($\Sigma_k$) and mean ($\mu_x$) for each class are estimated from the training data set (Eq.(3), Eq.(4)).

$$\mu_k = -\frac{1}{N_k} \sum_{i:y_i=k} x_i \qquad \text{Eq.(3)}$$

$$\text{Eq.(4)}$$
$$\Sigma_k = -\frac{1}{(N_k - 1)} \sum_{i:y_i=k} (x_i - \mu_k)^T (x_i - \mu_k)$$

Substituting a Gaussian density function for the $k^{th}$ class (Eq.(2) into Bayes theorem and taking the log values, the quadratic discriminant function is derived (Eq.(5)). Prior probabilities for class k ($\pi_k$) is calculated by the frequency of data points of class k in the training data (Eq.(6)). For a total number of $N$ points in the training observations, $N_k$ is the number of observations belong to kth class.

$$\delta_k(x) = -\frac{1}{2}(x - \mu_x)^T \Sigma_k^{-1}(x - \mu_x) + log\ \pi_k \qquad \text{Eq.(5)}$$

$$\pi_k = -\frac{N_k}{N} \qquad \text{Eq.(6)}$$

Covariance, mean and prior probability values are inserted into the discriminant function ($\delta_k(x)$) together with the state variables (Eq.(5)). The corresponding class is selected according to the largest value of the function. The number of parameters to be estimated for the QDA model for k classes and p predictors is $k.p.(p + 1)/2$. For this study, the QDA model output is the probability that an observation of a climate category will fall into each of the rainfall classes.

**2.3.2 Classification Tree model**

For the classification tree model the predictor space is divided into non-overlapping regions ($R_1..R_j$). A classification tree predicts each observation as belonging to the most commonly occurring class of the training data regions (James et.al., 2013). Recursive binary splitting is used to grow the classification tree.
Classification error rate, Gini index and cross-entropy are typically used to evaluate the quality of particular split (James et.al., 2013), and in our study we used the first two indices. Classification error rate ($E$) gives fraction of observation that do not belong to the most commonly occurring class of the training data regions (Eq.(7)). However, for the tree-growing, the Gini index ($G$) is considered as the criterion for splitting into regions (Eq.(8))

$$E = 1 - max_k(\hat{p}_{mk}) \qquad \text{Eq.(7)}$$

$$G = \sum_{k=1}^{K} \hat{p}_{mk}(1 - \hat{p}_{mk}) \qquad \text{Eq.(8)}$$

In Eq.(8) and Eq.(8) , $\hat{p}_{mk}$ represents the fraction of observations in the m[th] class that belong to the k[th] class. The Gini index is considered as a measure of node purity of the tree model, since small values of the index indicate that node has a higher number of observations from a single class.

The complexity of the trees are adjusted using a pruning process to produce more interpretable results. Complex trees reduces training error by overfitting the training data. Simple trees can be interpreted well, however, selecting a model which can find the pattern of data is important. In order to achieve the low classification error (training error + testing error), pruning technique is used. First, grow the very large tree, and sub tree is obtained by removing the weak links of the tree. Using tuning parameter to examine the trade-off between complexity of tree and the training error, and defining minimum samples for a node, maximum depth of the tree, and maximum number of terminal nodes are some of the pruning methods (Analytical Vidhya Team, 2016). For this study, we defined the maximum number of nodes to obtain the simple tree (pruned tree).

Tree models give the probability that an observation falls into each of the three rainfall classes. The predicted class is assigned based on the highest probability. Tree models handle ties of probability values by randomly assigning the class.

**2.3.3 Random Forest**

A random forest is an ensemble learning method used for classification and regression problems. The method is based on a multitude of decision trees based on training data with the final model as the mean of the ensemble (Breiman, 2001). Individual trees are built on a random sample of the training data with several predictors from the total number of predictors. Individual trees are built from the bootstrapped training data set.

There are some features, which can be tuned to make the better performed random forest model. Maximum number of predictors from the total predictors for individual trees, maximum number of trees, maximum node size of the trees and minimum sample leaf size are some of these features (Analytical Vidhya Team, 2015). In our study, we use the maximum number of trees as the main tuning parameters.

In a random forest model the importance of the variable is measured as the decrease in node impurity from the splits over the variable. This value is calculated by averaging the Gini index over the multitude of trees with a larger value indicating high importance of the predictor (James et.al., 2013).

**References:**

Analytical Vidhya Team. (2015). Tunning the parameters of your Random Forest model. Retrieved March 12, 2018, from https://www.analyticsvidhya.com/blog/2015/06/tuning-random-forest-model/

Analytical Vidhya Team. (2016). A Complete Tutorial on Tree Based Modeling from Scratch. Retrieved March 12, 2018, from https://www.analyticsvidhya.com/blog/2016/04/complete-tutorial-tree-based-modeling-scratch-in-python/

Breiman, L. (2001). Randomforest2001. *Machine Learning, 45(1)*, 5–32. https://doi.org/10.1017/CBO9781107415324.004

James, G., Witten, D., Hastie, T., & Tibshirani, R. (2013). *Springer Texts in Statistics An Introduction to Statistical Learning - with Applications in R*. https://doi.org/10.1007/978-1-4614-7138-7

Löwe, R., Madsen, H., & McSharry, P. (2016). Objective classification of rainfall in northern Europe for online operation of urban water systems based on clustering techniques. *Water (Switzerland)*, *8*(3). https://doi.org/10.3390/w8030087

---

## Author Response (AR1)

January 22, 2019

Re: Resubmission of manuscript Identifying ENSO Influences on Rainfall with Classification Models: Implications for Water Resource Management of Sri Lanka, Ms. No. hess-2018-249

Dr. Wouter Buytaert Editor, Hydrology and Earth System Sciences

Dear Dr. Buytaert,

Thank you for giving us the opportunity to revise our manuscript, *Identifying ENSO Influences* on *Rainfall with Classification Models: Implications for Water Resource Management of Sri Lanka.* We appreciate the careful review and constructive comments. We believe that the manuscript is improved after making the proposed edits to the introduction, methods, results, and discussion and after correcting writing mistakes.

Following this note are the reviewers comments in the blue color, followed by our comments explaining how and where the text was modified. Revisions were made in consultation with all the authors, and each author has approved the revised manuscript.

Thank you for your consideration.

Sincerely,

UTK ho Im

Thushara De Silva M.

**Referee #1**

General comments:

This study demonstrates application of different classification models to predict monthly rainfall using climate indices MEI and DMI. The findings of this study will be highly relevant for water managers of Sri Lanka. Below are my comments for Authors. Some of the similar comments I found that Authors have already addressed in the Discussion forum but please address all the points below.

**Response:** Thank you for identifying the importance of the study. Shortcomings of the paper identified by the reviewers will be rectified as suggested in the paper. Response to the detailed comments are given below.

**Specific comments:**

1) (a) Line 95 says that the river basin rainfall was calculated using the Thiessen polygon method. Why not divide the basin into sub-basins (using any GIS tool) based on digital elevation model and use sub-area averaged rainfall? Is this choice due to the fact that there are many reservoirs in the basin? Please clarify.

Response: We added the sentence below to clarify this point.

Since this study does not aim to explore rainfall across sub-basins, we do not use digital elevation maps to define the sub-basins.

(b) Out of 16 polygons in Mahaweli river basin and 11 in Kelani, what was the basis of selecting only 8 sub-basins?

**Response:** We revised the text as indicated below.

Considering the importance of sub-basins for the reservoir catchment and for water use, eight sub-basins are selected for analysis. Morape, Randenigala, Peradeniya, Manampitiya and Bowatenna represent the Mahaweli major reservoir catchments and irrigation tanks, and Norton Bridge, Norwood and Laxapana represent the Kelani basin reservoir catchments. The catchment of the major Mahaweli river reservoir cascade (Kotmale, Victoria, Randenigala, Rantambe, Bowatenna) is represented by Morape and Peradeniya located in the humid zone and by Randenigala and Bowatenna located in the intermediate zone. The arid zone major irrigation catchments of the Mahaweli are represented by Manampitiya. The catchment of the Kelaniya reservoir cascade (Norton Bridge & Moussakele) in the humid zone is represented by Laxapana, Norton Bridge and Norwood.

2) (a) Lines 104 to 109 describe how anomalies were calculated. Did you apply any of the transforms mentioned in line 108 to get normally distributed rainfall? Some plots/results can be included to clarify the rainfall anomaly classification.

**Response:** We added our response to this comment in the Appendix. We think that giving too many details in the manuscript decreases readability. Figure A.1 in the Appendix is referenced in the manuscript proper.

**@** Appendix**

**Normality Testing:**

The Shapiro-Wilk's method is used to identify the normality of rainfall anomaly distribution. The Manampitiya NEM normality test results are given below as an example.

Data 1: original data

W = 0.96675, p-value = 0.08185

Data 2: data transformed by square root

W = 0.98772, p-value = 0.7772

Data 3: data transformed by log

W = 0.91577, p-value = 0.0003325

Further, from data plots (Figure A. *1*) and the S-W statistic, we conclude that the square root transformed data is closer to being normally distributed than the other forms.

Figure A. 1: Manampitiya NEM standardized data (a) original form qqplot (b) square root form qqplot (c) original form density plot (d) square root form density plot

**(b) In Table1, the use of 0.5 appears like a random choice. Please justify.**

**Response: We explained in the Appendix; Figure A.2 is referenced in the paper.**

Extreme seasonal precipitation has been defined statistically in different ways using statistical thresholds (Easterling et al., 2000; Jentsch, Kreyling, & Beierkuhnlein, 2007; Knapp et al., 2015; Smith, 2011). We use 0.5 as a threshold to define three classes, which results in fairly evenly distributed data across the three classes (Figure A 2).

**(a) Appendix**

**Classification of data**

Using 0.5 as a threshold for a normal distribution defines portions of the data that are fairly evenly distributed into three categories – about 31%, 38%, and 31% for a normal distribution (Figure A 2). We deemed this a reasonable choice for our analysis.

3) (a) Line 122 says average of MEI and DMI were used but the figures 4 and 5 show that you have used them separately.

Response: We added below sentence to clarify the comment,

Because we analyzed the data in rainfall seasons, values of the climate indices over each season are averaged. For example for the NEM season, the MEI value is the average of January and February monthly values and for the SWM season, DMI is the average of May, June, July and September values.

(b) Authors should support the choice of MEI and DMI over several other climate indices which they could have used as predictors.

**Response:**

The paper was revised in several places and new results were added to the Appendix to support the choice of MEI and DMI climate indices.

**2.2 ENSO & IOD Indices**

The ENSO phenomenon is represented by MEI, NINO34, NINO3, NINO4 indices, and the Indian Ocean dipole phenomenon is represented by the DMI index. NINO34, NINO3, NINO4 indices are based on tropical sea surface temperature anomalies (National Center for Atmospheric Research, 2018) and the Multivariate ENSO Index (MEI) is based on sea-level pressure, zonal and meridional components of the surface wind, sea surface temperature, surface air temperature, and total cloudiness fraction of the sky (National Oceanic and Atmospheric Administration, 2017).

**4 Results**

Similar to other investigators, we observe several strong correlations between rainfall anomalies and the climate indices (Figure A 3, Table A 1, Appendix). Higher correlation values between MEI and rainfall anomalies can be seen compared to the correlation with other ENSO indices (Table A 1).

**(a)** Appendix**

Correlation analysis with multiple climate indices

We examined the correlation between rainfall anomalies and multiple climate indices to choose the two climate indices MEI and DMI ((Figure A 3, Table A 2). The ENSO phenomenon is represented by MEI, NINO34, NINO3, NINO4 indices. Correlation analysis indicates that MEI, which is estimated using several climate factors such as sea-level pressure, zonal and meridional components of the surface wind, sea surface temperature, surface air temperature, and total cloudiness fraction of the sky (National Oceanic and Atmospheric Administration, 2017), demonstrates higher correlation with rainfall anomalies in sub-basins for all rainfall seasons compared to the NINO34, NINO3 and NINO4. The Indian Ocean dipole phenomenon is represented by the DMI index, which represents the gradient of the sea surface temperature. Based on the correlation analysis and the content of the indices, we selected MEI as the indicator for ENSO and DMI as the indicator for IOD.

---

## Author Response (AR2)

February 24, 2019

Re: Resubmission of manuscript *Identifying ENSO Influences on Rainfall with Classification Models: Implications for Water Resource Management of Sri Lanka*, Ms. No. hess-2018-249

Dr. Wouter Buytaert
Editor, Hydrology and Earth System Sciences

Dear Dr. Buytaert,

Thank you for giving us the opportunity to revise our manuscript, *Identifying ENSO Influences on Rainfall with Classification Models: Implications for Water Resource Management of Sri Lanka.* We appreciate the careful review and constructive comments. We believe that the manuscript is improved after making the proposed edits to the figures.

Following this note are the editor's comments in the blue color, followed by our comments explaining how the figures were modified. Revisions were made in consultation with all the authors, and each author has approved the revised manuscript.

Thank you for your consideration.

Sincerely,

Thushara De Silva M.

Thank you for submitting your revised version of this manuscript. After careful reading of the revision, I think that the text is suitable for publication. However, the presentation, especially of the figures, suffers from some issues that can easily be addressed. In particular:

- Figure 1: the legend needs to be more precise: "Annual average precipitation [mm]". Also, the maps require a coordinate grid and scale.

The legend was revised as proposed and grid coordinates and scale were added to the map

- Figure 2: Explain in the caption the meaning of the abbreviations (NEM, FIM, SWM, SIM)

Figure 2 caption was revised as follows.

**Figure 1**: Sub basin Rainfall for (a) Morape, (b) Peradeniya,(c) Randenigala, (d) Bowatenna, (e) Laxapana (f) Norwood, (g) Norton Bridge, and (h) Manampitiya. Rainfall seasons are North East Monsoon (NEM), First Inter-Monsoon (FIM), South West Monsoon (SWM), and Second Inter-Monsoon (SIM)

- Figure 4: Using the "(" symbol is not ideal for readibility. Can you change this for instance to a cross or x?

Figure 4 "(" were removed, and instance with similar color code of the two other graphs (tree model) were included for the better readability.

- Figure 5: the text of this figure is very small - increase.

Figure 5 with increased font sized was included.

Once these issues are solved, I will happily accept the manuscript.

Thank you in advance for accepting the manuscript.

[revised manuscript text omitted]

**2.1    Sub-basin rainfall (Areal rainfall)**

Monthly rainfall data for years 1950-2013 are used for the study (Ceylon Electricity Board, 2017). River basin rainfall was calculated using the Thiessen polygon method (Viessman, 2002). The Mahaweli river basin is divided into 16 Thiessen polygons and the Kelani river basin is divided into 11 Thiessen polygons (Figure 1). Since this study does not aim to explore rainfall across sub-basins, we do not use digital elevation maps to define the sub-basins. Considering the importance of sub-basins for the reservoir catchment and for water use, eight sub-basins are selected for analysis. Morape, Randenigala, Peradeniya, Manampitiya and Bowatenna represent the Mahaweli major reservoir catchments and irrigation tanks, and Norton Bridge, Norwood and Laxapana represent the Kelani basin reservoir catchments. The catchment of the major Mahaweli river reservoir cascade (Kotmale, Victoria, Randenigala, Rantambe, Bowatenna) is represented by Morape and Peradeniya located in the humid zone and by Randenigala and Bowatenna located in the intermediate zone. The arid zone major irrigation catchments of the Mahaweli are represented by Manampitiya. The catchment of the Kelaniya reservoir cascade (Norton Bridge and Moussakele) in the humid zone is represented by Laxapana, Norton Bridge and Norwood.

We calculate the rainfall for the four seasons, NEM, FIM, SWM and SIM for 64 years of historical data. Rainfall anomalies are calculated by reducing the seasonal mean rainfall (Eq.(1)) and standardized anomalies are calculated by dividing the rainfall anomalies by the standard deviation (SD) (Eq.(2)).

$$X_{ANM} = (X - \bar{X}_t) \qquad \qquad \text{Eq.(1)}$$

$$X_{S\_ANM} = (X - \bar{X}_t)/SD_t \qquad \qquad \text{Eq.(2)}$$

Where, $\bar{X}_t$ is the average of seasonal rainfall, $X_{ANM}$ is the rainfall anomaly and $X_{S\_ANM}$ is the standardized rainfall anomaly.

Standardized rainfall anomalies are divided into three classes as dry, average and wet (Table 1). A normality test for the rainfall data classes is done using the Shapiro-Wilk test. If the rainfall data are not normally distributed, log (e), square root or square functions are used to transform the data into normally distributed data sets (Fig. A 1). Extreme seasonal precipitation has been defined statistically in different ways using statistical thresholds (Easterling et al.,

2000; Jentsch et.al., 2015; Smith, 2011). We use 0.5 as a threshold to define three classes, which results in fairly evenly distributed data across the three classes (Fig. A 2).

**Table 1** Rainfall anomaly classification

| Class | Range |
|-------|-------|
| dry | $X_{S\_ANM} < -0.5$ |
| average | $-0.5 <= X_{S\_ANM} < 0.5$ |
| wet | $0.5 <= X_{S\_ANM}$ |

**2.2    ENSO & IOD indices**

The ENSO phenomenon is represented by MEI, NINO34, NINO3, NINO4 indices, and the Indian Ocean dipole phenomenon is represented by DMI index. NINO34, NINO3, NINO4 indices are based on tropical sea surface temperature anomalies (National Center for Atmospheric Research, 2018) and the Multivariate ENSO Index (MEI) is based on sea-level pressure, zonal and meridional components of the surface wind, sea surface temperature, surface air temperature, and total cloudiness fraction of the sky (National Oceanic and Atmospheric Administration, 2017).

The Indian Ocean Dipole (IOD) is an oscillation of sea surface temperature in the equatorial Indian ocean between

Arabian sea and south of Indonesia (Bureau of Meteorology Australia, 2017). IOD is identified as relevant to the climate of Australia (Power et.al., 1999) and countries surrounded by the Indian ocean in southern Asia (Chaudhari et al., 2013; Maity and Nagesh Kumar, 2006; Qiu et al., 2014; Surendran et al., 2015). The Dipole Mode Index (DMI)

is used to represent the IOD capturing the west and eastern equatorial sea surface temperature gradient.

Data used for the analyses are NINO34, NINO3, NINO4, MEI monthly data from years 1950 – 2013, (National

Oceanic and Atmospheric Administration, 2017; National Center for Atmospheric Research, 2018), and the DMI

monthly data from years 1950-2013 ( HadISST dataset, Japan Agency for Marine-Earth Science and Technology

2017). Because we analyzed the data in rainfall seasons, values of the climate indices over the season are averaged.

For example for the NEM season, the MEI value is the average of January and February monthly values and for the

SWM season, DMI is the average of May, June, July and September values.

**3    Methods**

Seasonal values of MEI and DMI were used as the predictors to classify seasons into the three rainfall classes. The total data set is divided into 75 % for training the model and 25 % for testing model performance. Quadratic discriminant analysis (QDA) and classification trees were selected for the analyses. A random forest model also was applied to investigate the reliability of a cross-validated statistical forecast tool based on an advance estimate of MEI

and DMI. We used the R programming language to carry out the statistical analyses. R packages: caret, tree, randomForest, fitdistriplus, devtools and quantreg are used for the studies.

**3.1    Quadratic Discriminant Analysis (QDA)**

The mathematical formulation of QDA can be derived from Bayes theorem assuming that observations from each class are drawn from a Gaussian distribution (James et.al., 2013; Löwe et.al., 2016).

The prior probability $\pi_k$ represents the randomly chosen observation coming from kth class with density function $f_k(x)$. Bayes theorem states that

$$Pr(Y = k|X = x) = -\frac{\pi_k f_k(x)}{\sum_{l=1}^{K} \pi_l f_l(x)} \qquad \text{Eq.(3)}$$

In Eq.(3), the posterior probability $Pr(Y = k|X = x)$ indicates that observation $X = x$ belongs to the kth class. For p predictors, the multivariate Gaussian distribution density function is defined for every class k (Eq.(4)).

$$f_k(x) = -\frac{1}{(2\pi)^{p/2}|\sum_k|^{1/2}} \, exp\left(-\frac{1}{2}(x - \mu_k)^T \Sigma_k^{-1}(x - \mu_k)\right) \qquad \text{Eq.(4)}$$

In Eq.(2), $\sum_k$ is the covariance matrix and $\mu_x$ is the mean vector. The covariance matrix ($\sum_k$) and mean ($\mu_x$) for each class are estimated from the training data set (Eq.(5), Eq.(6)).

$$\mu_k = -\frac{1}{N_k} \sum_{i:y_i=k} x_i \qquad \text{Eq.(5)}$$

$$\Sigma_k = -\frac{1}{(N_k - 1)} \sum_{i:y_i=k} (x_i - \mu_k)^T (x_i - \mu_k) \qquad \text{Eq.(6)}$$

Substituting a Gaussian density function for the k[th] class (Eq.(4)) into Bayes theorem and taking the log values, the quadratic discriminant function is derived (Eq.(7)). Prior probabilities for class k ($\pi_k$) is calculated by the frequency of data points of class k in the training data (Eq.(8)). For a total number of $N$ points in the training observations, $N_k$ is the number of observations belong to kth class.

$$\delta_k(x) = -\frac{1}{2}(x - \mu_x)^T \Sigma_k^{-1}(x - \mu_x) + log \, \pi_k \qquad \text{Eq.(7)}$$

$$\pi_k = -\frac{N_k}{N} \qquad \text{Eq.(8)}$$

Covariance, mean and prior probability values are inserted into the discriminant function ($\delta_k(x)$) together with the state variables (Eq.(5)). The corresponding class is selected according to the largest value of the function. The number of parameters to be estimated for the QDA model for k classes and p predictors is $k.p.(p + 1)/2$. For this study, the

QDA model output is the probability that an observation of a climate category will fall into each of the rainfall classes.

**3.2 Classification Tree model**

For the classification tree model the predictor space is divided into non-overlapping regions ($R_1..R_j$). A classification tree predicts each observation as belonging to the most commonly occurring class of the training data regions (James et.al., 2013). Recursive binary splitting is used to grow the classification tree.

Classification error rate, Gini index and cross-entropy are typically used to evaluate the quality of particular split (James et.al., 2013), and in our study we used the first two indices. Classification error rate ($E$) gives fraction of observation that do not belong to the most commonly occurring class of the training data regions (Eq.(9)). However, for the tree-growing, the Gini index ($G$) is considered as the criterion for splitting into regions (Eq.(10))

$$E = 1 - max_k(\hat{p}_{mk}) \qquad\qquad \text{Eq.(9)}$$

$$G = \sum_{k=1}^{K} \hat{p}_{mk}(1 - \hat{p}_{mk}) \qquad\qquad \text{Eq.(10)}$$

In Eq.(9) and Eq.(10), $\hat{p}_{mk}$ represents the fraction of observations in the m[th] class that belong to the k[th] class. The Gini index is considered as a measure of node purity of the tree model, since small values of the index indicate that node has a higher number of observations from a single class.

The complexity of the trees are adjusted using a pruning process to produce more interpretable results. Complex trees reduces training error by overfitting the training data. Simple trees can be interpreted well, however, selecting a model which can find the pattern of data is important. In order to achieve the low classification error (training error + testing error), pruning technique is used. First, grow the very large tree, and sub tree is obtained by removing the weak links of the tree. Using tuning parameter to examine the trade-off between complexity of tree and the training error, and defining minimum samples for a node, maximum depth of the tree, and maximum number of terminal nodes are some of the pruning methods (Analytical Vidhya Team, 2016). For this study, we defined the maximum number of nodes to obtain the simple tree (pruned tree).

Tree models give the probability that an observation falls into each of the three rainfall classes. The predicted class is assigned based on the highest probability. Tree models handle ties of probability values by randomly assigning the class.

**3.3 Random Forest**

A random forest is an ensemble learning method used for classification and regression problems. The method is based on a multitude of decision trees based on training data with the final model as the mean of the ensemble (Breiman, 2001). Individual trees are built on a random sample of the training data with several predictors from the total number of predictors. Individual trees are built from the bootstrapped training data set.

There are some features, which can be tuned to make the better performed random forest model. Maximum number of predictors from the total predictors for individual trees, maximum number of trees, maximum node size of the trees and minimum sample leaf size are some of these features (Analytical Vidhya Team, 2015). In our study, we use the maximum number of trees as the main tuning parameters.

In a random forest model the importance of the variable is measured as the decrease in node impurity from the splits
over the variable. This value is calculated by averaging the Gini index over the multitude of trees with a larger value
indicating high importance of the predictor (James et.al., 2013).

[Figure]

**Figure 2**: Sub basin Rainfall for (a) Morape, (b) Peradeniya,(c) Randenigala, (d) Bowatenna, (e) Laxapana (f) Norwood, (g) Norton Bridge, and (h) Manampitiya. Rainfall seasons are North East Monsoon (NEM), First Inter-Monsoon (FIM), South West Monsoon (SWM), and Second Inter-Monsoon (SIM)

**4    Results**

Monthly rainfall boxplots of eight sub basins over the year for 1950 - 2013 illustrate the seasonal and the spatial variation of rainfall patterns (Fig. 2). The largest fraction of total rainfall in the arid zone occurs at the end of the SIM

(December) and during the NEM (January - February) with correspondingly high variability whereas there is little rainfall in the arid zone during the SWM (May - September) with correspondingly little variability (Fig. 2 (h)). The intermediate zone receives approximately 60% of total rainfall from the SIM and NEM. Although the variability of the rainfall is low in the intermediate zone, high rainfall can occur in all seasons (Fig. 2 (c) and (d)). In the humid zone, a large portion of rainfall occurs in SWM and early months of SIM (October-November). High variability of humid zone rainfall is observed at the end of FIM (April), in the SWM (May-September), and at the start of SIM

(October) (Fig. 2 (a), (b), (e), (f) and (g)).

Similar to other investigators, we observe several strong correlations between rainfall anomalies and the climate indices (Table A. 1, Table A. 2, and Appendix). Higher correlation values between MEI and rainfall anomalies can be seen compared to the correlation with other ENSO indices (Table A. 1).  In addition, rainfall in the SWM is very important for stations in the humid zone of the country which is the source of a large amount of water stored in reservoirs (Table A. 2). Correlation coefficients between SWM rainfall at Norton Bridge are negative and strong, -

0.31 for MEI (p = 0.01) and -0.37 for DMI (p < 0.01). The strength of the correlation notwithstanding, the residuals from a regression model indicate that high uncertainty would attach to any forecast (Fig. 3). Thus, we are led to explore the efficacy of classification methods (Appendix).

[Figure]

**Figure 3** Linear regression of rainfall anomaly on MEI and DMI. High values of MEI and DMI are associated with
low values of rainfall.

We present classification results for two sub-basins, one that has the highest rainfall during the NEM, Manampitiya, and one that has the highest rainfall for the SWM, Norton Bridge (Fig. 4). Norton Bridge represents the areal rainfall of reservoir catchments in the wet zone and Manampitiya represents the rainfall that contributes to irrigation tanks in the dry zone. Results of other sub-basins are presented in the supplementary materials (Fig. A 4, Fig. A 5, Fig. A 6,

Fig. A 7, Appendix). Because MEI has higher correlation with rainfall anomalies than other ENSO indices, classification was done with only MEI and DMI.

The SWM is a season when the humid zone receives the bulk of rainfall. At Norton Bridge, the occurrences of the dry
rainfall anomaly class in the SWM is seen to "clump" in the region of relatively high MEI and DMI. Both the
classification tree and the QDA successfully identify the pattern (Fig. 4 (a) and (c)) with an overall accuracy of 73 %,
19 and 16 correct out of 22 occurrences (Table 2). In the arid zone the NEM season is one of the most important for
rainfall. At Manampitiya, the MEI provides the primary variable in the classification, with the dry anomaly class being
correctly selected in 52 % by tree model and 95 % with the QDA model. The results suggest that it may be possible
to identify seasons when it is expected to be anomalously dry. The correct classification of "average" conditions likely
has less importance for water managers. We explored classification using two classes, "Dry" and "Not Dry." In this
case, the classification model again correctly classifies 86 % of the anonymously dry cases and gets more than 69 %
of the "Not Dry" cases correct (Fig. 5).

[Figure]

**Figure 4** Norton Bridge and Manampitiya rainfall classes (dry, average, wet) identified by ENSO and IOD
phenomena. (a) Norton Bridge SWM rainfall classification tree model (b) Manampitiya NEM rainfall classification
tree model (c) Norton Bridge SWM rainfall QDA (d) Manampitiya NEM rainfall classification by QDA

[Figure]

**Figure 5** Classification tree for Norton Bridge SWM rainfall using two categories (dry and not dry)

**Table 2** Classification model results. Highlighted cells indicate where there may be information content with respect
to forecasting either dry or wet anomaly classes as judged by a classification success rate of at least 2/3.

| Season | Manampitiya | | | Norton Bridge | | |
|---|---|---|---|---|---|---|
| | QDA Model | | | QDA Model | | |
| | Dry | Normal | Wet | Dry | Normal | Wet |
| NEM | 22/23 | 11/25 | 1/16 | 5/20 | 25/29 | 2/15 |
| FIM | 9/21 | 20/24 | 5/19 | 3/20 | 14/23 | 14/20 |
| SWM | 2/21 | 30/27 | 2/16 | 16/22 | 9/22 | 9/20 |
| SIM | 17/25 | 13/20 | 7/19 | 7/22 | 15/22 | 11/20 |
| Season | Tree Model | | | Tree Model | | |
| | Dry | Normal | Wet | Dry | Normal | Wet |
| NEM | 12/23 | 9/25 | 11/16 | 11/20 | 18/29 | 8/15 |
| FIM | 9/21 | 19/24 | 8/19 | 13/21 | 6/23 | 15/20 |
| SWM | 6/21 | 25/27 | 7/16 | 19/22 | 8/22 | 9/20 |
| SIM | 20/25 | 0/20 | 17/19 | 19/22 | 5/22 | 14/20 |

Classification trees are known to be unstable. That is, small changes in the observations can lead to large changes in
the decision tree. The random forest approach overcomes the issue by building a "bag" of trees from bootstrap samples.
The robustness of the model can then be checked by considering the "out-of-bag" error. The results of the random
forest indicate that predictions of three rainfall anomaly classes using MEI and DMI is not feasible (Table 3). The out-
of-bag error rate is close to two thirds, which for three categories is equivalent to a random selection.

**Table 3** Results of random forest ensemble classification results

| Season | Norton Bridge | | | | Manampitiya | | | |
|---|---|---|---|---|---|---|---|---|
| | Dry | Normal | Wet | OOB Er | Dry | Normal | Wet | OOB Er |
| NEM | 11/20 | 12/29 | 6/15 | 55% | 14/23 | 10/25 | 5/16 | 55% |

| FIM | 7/21 | 8/23 | 8/20 | 64% | 10/21 | 11/24 | 6/19 | 58% |
|---|---|---|---|---|---|---|---|---|
| SWM | 9/22 | 6/22 | 8/20 | 64% | 6/21 | 17/27 | 5/16 | 56% |
| SIM | 13/22 | 9/22 | 9/20 | 52% | 15/25 | 8/20 | 7/19 | 53% |

However, the results of the random forest for a classification as either "Dry" or "Not Dry" suggests that there may be skill in such a prediction. The out-of-bag error rates for this case range from 22 % to 38 % for Norton Bridge and Manampitiya (Table 3) and from 20 % to 39 % across all stations (Table A. 7).

**Table 4** Results of random forest ensemble classification results for two rainfall anomaly classes

| Season | Norton Bridge | | | Manampitiya | | |
|---|---|---|---|---|---|---|
| | Dry | Not dry | OOB Error | Dry | Not dry | OOB Error |
| NEM | 9/20 | 36/44 | 30 % | 13/23 | 33/41 | 28 % |
| FIM | 5/21 | 35/43 | 38 % | 8/21 | 35/43 | 33 % |
| SWM | 9/22 | 32/42 | 36 % | 5/16 | 34/43 | 39 % |
| SIM | 10/22 | 36/42 | 28 % | 16/25 | 34/39 | 22 % |

The QDA method produces results that are promising with respect to identification of extreme dry events as indicated by seasonal rainfall (Table 5).

**Table 5** Classification results for extreme dry (very low rainfall) and wet (very high rainfall) seasons.

| Class | Range | NortonBridge SWM | | Manampitiya NEM | |
|---|---|---|---|---|---|
| | | tree | QDA | tree | QDA |
| Very dry | $X_{S\_ANM} < -1.0$ | 10/11 | 10/11 | 6/11 | 11/11 |
| dry | $-1.0 <= X_{S\_ANM} < -0.5$ | 9/11 | 6/11 | 5/11 | 9/10 |
| average | $-0.5 <= X_{S\_ANM} < 0.5$ | 8/22 | 9/22 | 9/25 | 11/25 |
| wet | $0.5 <= X_{S\_ANM} <= 1.0$ | 5/11 | 5/11 | 1/5 | 0/5 |
| Very wet | $1.0 <= X_{S\_ANM}$ | 6/11 | 6/11 | 7/11 | 1/11 |

**5    Discussion**

Understanding seasonal rainfall variability across the spatially diverse Mahaweli and Kelani river basins is important for irrigation and hydropower water planning. SWM and SIM are the key rainfall seasons for sub basins in the humid zone (Norton Bridge, Morape, Peradeniya and Laxapana), delivering 80 % of annual rainfall (Fig. 2 (a),(b),(e),(f)). For the arid zone (Manampitiya) and intermediate zone (Randenigala, Bowatenna) sub basins, the major season is SIM, which delivers more than 40 % of annual rainfall (Fig. 2 (c),(d),(h)). The arid zone also gets rainfall during the NEM (24 % of annual rainfall at Manampitiya) and the intermediate zone gets rainfall during the SWM (25 % - 30 % of annual rainfall at Randenigala and Bowatenna).

Climate teleconnection indices are related to rainfall anomalies observed during the two main growing seasons, Yala and Maha. The Maha agriculture season (October-March) depends on rain from SIM and NEM. During El Nino events rainfall increases for the first three months of the Maha season (SIM: October-December) (Fig. A 4, Fig. A 5, Fig. A

6, Fig. A 8) (Ropelewski and Halpert, 1995) and decreases during the last three months (NEM: January-March)(Fig.

4 (b)). In Yala season (April-September), La-Nina events enhance the rainfall during SWM (Fig. 4(a), (c), Fig. A 4,

Fig. A 5, Fig. A 6, Fig. A 8)(Whitaker et.al, 2001). During El Nino events the SWM rainfall is reduced (Fig. 4 (a), (c),

Fig. A 8, Fig. A 9) (Chandrasekara et.al., 2017; Chandimala and Zubair, 2007; Zubair, 2003). The El Nino impact during the SWM is not as significant as it is during the NEM season (International Research Institute, 2017a). We find, however, that there is an interaction between two teleconnection indices, MEI and IOD for SWM rainfall. During the Yala season there is a high probability of having a drought when both the IOD and MEI are positive (Fig. 5). Also not having drought is probable when both the IOD and MEI are negative (Fig. 5, Fig. A 8, Fig. A 9).

Classification of wet, average, and dry rainfall anomalies using the MEI and DMI indices is successful. For example, a dry SWM season for Norton Bridge (Table 2) and other humid-zone stations (Table A. 4) is classified correctly with greater than 70 % accuracy with QDA and tree models. However, a random forest approach demonstrates that there is little skill in identifying a full wet-average-dry classification. However, a random forest model using only two rainfall categories shows more than 60 % accuracy in identifying "dry" and "not dry" classes of key rainfall seasons of the humid zone (Table 4, Table A. 7). Similarly, for arid zone locations such as Manampitiya, the dry rainfall class identification for NEM and SIM seasons is about 60 % (Table 4, Table A. 7).

Our statistical classification models can be combined with MEI and DMI forecasts to indicate the season-ahead expectation for rainfall. ENSO forecasts are available from the International Research Institute for Climate and Society (International Research Institute, 2017b) and IOD forecasts are available in the Bureau of Meteorology (BOM),

Australian Government (Bureau of Meteorology, 2017). ENSO and IOD predictions are also associated with the uncertainty. Therefore, final forecast accuracy is a combination of the MEI, DMI forecast uncertainties and model's accuracy rate in each class. Although overall prediction accuracy is not extremely high, a forecast of an anomalously low rainfall season can have value for risk-averse farmers (Cabrera et.al., 2007) and can guide plans for hydropower management (Block and Goddard, 2012).

The electricity and agriculture sectors of Sri Lanka heavily rely on Mahaweli and Kelani river water resources so season ahead forecasts of abnormally low rainfall should be useful for decisions on adaptation measures. For example, water availability of the first three months of a growing season is important for crop selection and the extent of land to be cultivated. Hydropower planning and scheduling of maintenance of the power plants also can benefit from season-ahead forecasts. The damage that can occur due to incorrect rainfall forecasts in the agriculture and energy sectors can be minimized with emergency planning during the season, which is the usual practice.

Although the accuracy of predicting low or not low seasonal rainfall is not very high, decisions based on forecasts that are improvements over climate averages should be an improvement over current practices. The accuracy of statistical models can be improved with longer records, which are important to train the classification models. Also, models can be fine-tuned for important shorter periods such as crop planting months and harvesting months for irrigation water planning.

**6    Conclusion**

ENSO and IOD phenomena teleconnections with river basin rainfall provide potentially useful information for water resource management. Relationships identified between teleconnection indices and river basin rainfall agree with other research findings. Prediction of seasonal rainfall classes from ENSO and IOD indices can inform water resources managers in reservoir operation planning for both hydropower and irrigation releases.

**Author contributions.**

TDM and GMH conceptualized the study and TDM carried out the data analysis. TDM prepared the paper with contribution from GMH.

**Acknowledgement.**

This research is part of a multidisciplinary research initiative called Agricultural Decision-Making and Adaptation to

Precipitation Trends in Sri Lanka (ADAPT-SL) at Vanderbilt Institute for Energy and Environment (VIEE). The work is supported by WSC Program Grant No. NSF-EAR 1204685.

**Appendix: Identifying ENSO Influences on Rainfall with Classification Models: Implications for Water Resource Management of Sri Lanka**

**Normality Testing:**

The Shapiro-Wilk's method is used to identify the normality of rainfall anomaly distribution. The Manampitiya NEM

normality test results are given below as an example.

Data 1: original data

W =  0.96675, p-value = 0.08185

Data 2: data transformed by square root

W =  0.98772, p-value = 0.7772

Data 3: data transformed by log

W =  0.91577, p-value = 0.0003325

Further, from data plots (Fig. A 1) and the S-W statistic, we conclude that the square root transformed data is closer to being normally distributed than the other forms.

[Figure]

**Figure A 1** Manampitiya NEM standardized data (a) original form qqplot (b) square root form qqplot (c) original form density plot (d) square root form density plot

**Classification of data**

Using 0.5 as a threshold for a normal distribution defines portions of the data that are fairly evenly distributed into three categories – about 31%, 38%, and 31% for a normal distribution (Fig. A 2). We deemed this a reasonable choice for our analysis.

[Figure]

**Figure A 2** (a) Norton Bridge SWM rainfall anomaly distribution (b) Manampitiya NEM rainfall anomaly distribution

**Correlation analysis with multiple climate indices**

We examined the correlation between rainfall anomalies and multiple climate indices to choose the two climate indices MEI and DMI (Fig. A 3, Table A. 1). The ENSO phenomenon is represented by MEI, NINO34, NINO3, NINO4 indices. Correlation analysis indicates that MEI, which is estimated using several climate factors such as sea-level pressure, zonal and meridional components of the surface wind, sea surface temperature, surface air temperature, and total cloudiness fraction of the sky (National Oceanic and Atmospheric Administration, 2017), demonstrates higher correlation with rainfall anomalies in sub-basins for all rainfall seasons compared to the NINO34, NINO3 and NINO4. The Indian Ocean dipole phenomenon is represented by the DMI index, which represents the gradient of the sea surface temperature. Based on the correlation analysis and the content of the indices, we selected MEI as the indicator for ENSO and DMI as the indicator for IOD.

[Figure]

**Figure A 3** Correlation between Norwood rainfall anomalies with multiple climate indices

**Table A. 1** Correlation analysis of rainfall anomalies and climate indices

| Rainfall | Morape | | | | | Peradeniya | | | | |
|---|---|---|---|---|---|---|---|---|---|---|
| Month | MEI | NINO34 | NINO3 | NINO4 | *DMI* | MEI | NINO34 | NINO3 | NINO4 | *DMI* |
| NEM | -0.35 | -0.35 | -0.34 | -0.38 | *-0.09* | -0.38 | -0.40 | -0.39 | -0.42 | *-0.11* |
| FIM | -0.28 | -0.19 | -0.28 | -0.07 | *-0.11* | -0.27 | -0.18 | -0.30 | -0.06 | *-0.06* |
| SWM | -0.35 | -0.24 | -0.23 | -0.26 | *-0.29* | -0.35 | -0.26 | -0.25 | -0.27 | *-0.31* |
| SIM | 0.21 | 0.23 | 0.27 | 0.19 | *0.12* | 0.17 | 0.19 | 0.21 | 0.15 | *0.09* |

| Rainfall | Laxapana | | | | | Norwood | | | | |
|---|---|---|---|---|---|---|---|---|---|---|
| Month | MEI | NINO34 | NINO3 | NINO4 | *DMI* | MEI | NINO34 | NINO3 | NINO4 | *DMI* |
| NEM | -0.27 | -0.26 | -0.28 | -0.27 | *-0.01* | -0.28 | -0.26 | -0.29 | -0.27 | *-0.04* |
| FIM | -0.28 | -0.16 | -0.27 | -0.03 | *-0.07* | -0.27 | -0.18 | -0.26 | -0.03 | *-0.13* |
| SWM | -0.3 | -0.23 | -0.21 | -0.25 | *-0.31* | -0.21 | -0.12 | -0.15 | -0.16 | *-0.24* |
| SIM | 0.1 | 0.10 | 0.14 | 0.06 | *0.08* | 0.29 | 0.31 | 0.32 | 0.27 | *0.28* |

| Rainfall | Randenigala | | | | | Bowatenna | | | | |
|---|---|---|---|---|---|---|---|---|---|---|
| Month | MEI | NINO34 | NINO3 | NINO4 | *DMI* | MEI | NINO34 | NINO3 | NINO4 | *DMI* |
| NEM | -0.30 | -0.31 | -0.29 | -0.34 | *-0.11* | -0.35 | -0.36 | -0.35 | -0.38 | *-0.2* |
| FIM | -0.29 | -0.23 | -0.33 | -0.10 | *-0.04* | -0.23 | -0.17 | -0.25 | -0.09 | *-0.02* |
| SWM | -0.17 | -0.12 | -0.09 | -0.18 | *-0.24* | -0.18 | -0.09 | -0.05 | -0.11 | *-0.12* |
| SIM | 0.37 | 0.38 | 0.41 | 0.36 | *0.35* | 0.35 | 0.41 | 0.40 | 0.40 | *0.36* |

| Rainfall | Norton Bridge | | | | | Manampitiya | | | | |
|---|---|---|---|---|---|---|---|---|---|---|
| Month | MEI | NINO34 | NINO3 | NINO4 | *DMI* | MEI | NINO34 | NINO3 | NINO4 | *DMI* |
| NEM | -0.32 | -0.30 | -0.33 | -0.33 | *-0.01* | -0.26 | -0.28 | -0.26 | -0.28 | *-0.16* |
| FIM | -0.18 | -0.12 | -0.21 | -0.01 | *-0.08* | -0.2 | -0.17 | -0.31 | -0.06 | *-0.14* |
| SWM | -0.31 | -0.22 | -0.21 | -0.22 | *-0.37* | -0.07 | 0.08 | 0.08 | -0.01 | *-0.03* |
| SIM | 0.02 | -0.02 | 0.03 | -0.04 | *-0.15* | 0.45 | 0.46 | 0.44 | 0.46 | *0.51* |

**Correlation analysis with MEI and DMI climate indices**

Correlation coefficients between rainfall anomalies and MEI and DMI are negative for the NEM, FIM and SWM
seasons and positive for the SIM season. Rainfall anomalies correlations to the DMI are not stronger as the correlations
to the MEI. However, there are strong correlations for the anomalies of major monsoons to the sub basins and DMI
values. For example, wet sub basins (Morape, Peradeniya, Laxapana, Norwood, Norton Bridge) have high correlation
coefficient between SWM rainfall anomalies and DMI, while dry zone (Manampitiya) and intermediate zone
(Randenigala, Bowatenna) sub basins have high correlation coefficient between NEM and SIM rainfall anomalies.

**Table A. 2** Correlation between rainfall anomalies and MEI, DMI indices. High correlation coefficients are
highlighted.

| Rainfall | Morape | | Peradeniya | | Randenigala | | Bowatenna | |
|---|---|---|---|---|---|---|---|---|
| Month | MEI | DMI | MEI | DMI | MEI | DMI | MEI | DMI |
| NEM | **-0.35** | -0.09 | **-0.38** | -0.11 | **-0.30** | -0.11 | **-0.35** | **-0.20** |

| | MEI | DMI | MEI | DMI | MEI | DMI | MEI | DMI |
|---|---|---|---|---|---|---|---|---|
| FIM | **-0.28** | -0.11 | **-0.27** | -0.06 | **-0.29** | -0.04 | **-0.23** | -0.02 |
| SWM | **-0.35** | **-0.29** | **-0.35** | **-0.31** | -0.17 | **-0.24** | -0.18 | -0.12 |
| SIM | **0.21** | 0.12 | 0.17 | 0.09 | **0.37** | **0.35** | **0.35** | **0.36** |

| Rainfall | Laxapana | | Norwood | | Norton Bridge | | Manampitiya | |
|---|---|---|---|---|---|---|---|---|
| Month | MEI | DMI | MEI | DMI | MEI | DMI | MEI | DMI |
| NEM | **-0.27** | -0.01 | **-0.28** | -0.04 | **-0.32** | -0.01 | **-0.26** | **-0.16** |
| FIM | **-0.28** | -0.07 | **-0.27** | -0.13 | **-0.18** | -0.08 | **-0.20** | -0.14 |
| SWM | **-0.30** | **-0.31** | **-0.21** | **-0.24** | **-0.31** | **-0.37** | -0.07 | -0.03 |
| SIM | 0.10 | 0.08 | **0.29** | **0.28** | 0.02 | -0.15 | **0.45** | **0.51** |

Classification methods classification tree models, random forest and quadratic discriminant analysis identify the relationship between standardized rainfall anomaly classes (dry, average, wet) and MEI and DMI values (Fig. A 4, Fig. A 5, Fig. A 6, Fig. A 7). Positive values of MEI and DMI values resulted dry or average rainfall class for the NEM, FIM and SWM seasons. However, for SIM rainfall has wet or average class for the positive values of MEI and DMI. Accuracy of model result are high for the dominant monsoon rainfall seasons of each sub basin (Table A. 3, Table A. 4, Table A. 5). Ensemble model approach with random forest has given comparatively lower out-of-bag error rate for the dominant monsoons' rainfall anomaly classification (Table A. 5). For example, wet zone sub basins such as Norton Bridge, Norwood, Laxapana, Peradeniya and Morape random forest error rate is lower for the SWM and SIM seasons. Same as, dry and intermediate sub basins Manampitiya, Randenigala and Bowatenna NEM and SIM rainfall classes accuracy rate is high than other rainfall seasons. Also all three models have higher accuracy rate in identifying dry events and error rate of identifying wet and dry class also less 15% (Table A. 3, Table A. 4, Table A. 5). Further analysis of two rainfall classes dry and not dry rainfall classes are identified relevant to the MEI and DMI values with classification tree and random forest methods (Fig. A 8, Fig. A 9). Classification tree models for two classes have higher accuracy rate as 65% - 84% for eight sub basins (Table A. 6). Random forest out-of-bag error for two classes models are vary between 20% - 39% and shows higher skill in identifying rainfall classes for major monsoons of the sub basins (Table A. 7). MEI shows higher variable importance of identifying the rainfall classes compare to the DMI values. Specially, for NEM and SIM which are important to the dry zone sub basins importance of MEI is high in the classification. However, some of the wet zone sub basins shows equal importance of DMI variable in identifying two rainfall classes in FIM and SWM (Fig. A 10).

[Figure]

**Figure A 4** Identifying relationships between three rainfall classes (dry, average, wet) and MEI and DMI values using classification tree models.(a)Morape (b)Peradeniya (c)Randenigala (d)Bowatenna

[Figure]

**Figure A 5** Identifying relationships between three rainfall classes (dry, average, wet) and MEI and DMI values using classification tree models. (e)Laxapana (f)Norwood (g)Norton Bridge (h)Manampitiya

**Table A. 3** Classification tree model results. Highlighted cells indicate where there may be information content with
respect to forecasting either dry or wet anomaly classes

| Season | Morape | | | Peradeniya | | |
|---|---|---|---|---|---|---|
| | Dry | Normal | Wet | Dry | Normal | Wet |
| NEM | 21/21 | 13/29 | 0/14 | 10/20 | 24/31 | 0/13 |
| FIM | 5/19 | 19/25 | 12/20 | 5/20 | 28/28 | 6/16 |
| SWM | 12/24 | 13/21 | 12/19 | 9/23 | 11/19 | 18/22 |
| SIM | 8/19 | 18/28 | 9/17 | 12/25 | 16/19 | 5/20 |
| Season | Randenigala | | | Bowatenna | | |
| | Dry | Normal | Wet | Dry | Normal | Wet |
| NEM | 11/24 | 11/25 | 12/15 | 24/24 | 12/19 | 0/21 |
| FIM | 8/20 | 24/25 | 3/19 | 17/21 | 17/25 | 0/18 |
| SWM | 8/21 | 23/24 | 8/19 | 18/25 | 6/21 | 12/18 |
| SIM | 14/24 | 11/21 | 15/19 | 17/21 | 9/26 | 13/17 |
| Season | Laxapana | | | Norwood | | |
| | Dry | Normal | Wet | Dry | Normal | Wet |
| NEM | 0/19 | 24/24 | 6/21 | 4/19 | 22/28 | 10/17 |
| FIM | 2/20 | 14/26 | 18/18 | 7/19 | 19/21 | 12/24 |
| SWM | 19/23 | 14/20 | 8/21 | 10/20 | 14/27 | 11/17 |
| SIM | 8/21 | 22/26 | 9/17 | 16/20 | 15/25 | 11/19 |
| Season | Norton Bridge | | | Manampitiya | | |
| | Dry | Normal | Wet | Dry | Normal | Wet |
| NEM | 11/20 | 18/29 | 8/15 | 12/23 | 9/25 | 11/16 |
| FIM | 13/21 | 6/23 | 15/20 | 9/21 | 19/24 | 8/19 |
| SWM | 19/22 | 8/22 | 9/20 | 6/21 | 25/27 | 7/16 |
| SIM | 19/22 | 5/22 | 14/20 | 20/25 | 0/20 | 17/19 |

[Figure]

**Figure A 6** Identifying relationships between three rainfall classes (dry, average, wet) and MEI and DMI values
using QDA models.(a) Morape (b) Peradeniya (c) Randenigala (d) Bowatenna

[Figure]

**Figure A 7** Identifying relationships between three rainfall classes (dry, average, wet) and MEI and DMI values using classification tree models. (e) Laxapana (f) Norwood (g) Norton Bridge (h) Manampitiya

**Table A. 4** Classification QDA model results. Highlighted cells indicate where there may be information content
with respect to forecasting either dry or wet anomaly classes

| Season | Morape | | | Peradeniya | | |
|---|---|---|---|---|---|---|
| | Dry | Normal | Wet | Dry | Normal | Wet |
| NEM | 6/21 | 28/29 | 0/14 | 10/20 | 28/31 | 0/13 |
| FIM | 7/19 | 22/25 | 9/20 | 5/20 | 28/28 | 2/16 |
| SWM | 19/24 | 6/21 | 13/19 | 20/23 | 6/19 | 13/22 |
| SIM | 5/19 | 26/28 | 2/17 | 13/25 | 16/19 | 4/20 |
| Season | Randenigala | | | Bowatenna | | |
| | Dry | Normal | Wet | Dry | Normal | Wet |
| NEM | 17/24 | 8/25 | 4/15 | 24/24 | 9/19 | 3/21 |
| FIM | 8/20 | 13/25 | 12/19 | 9/21 | 23/25 | 1/18 |
| SWM | 4/21 | 13/24 | 8/19 | 19/25 | 7/21 | 8/18 |
| SIM | 19/24 | 16/21 | 6/19 | 13/21 | 15/26 | 10/17 |
| Season | Laxapana | | | Norwood | | |
| | Dry | Normal | Wet | Dry | Normal | Wet |
| NEM | 4/19 | 15/24 | 14/21 | 8/19 | 23/28 | 6/17 |
| FIM | 4/20 | 22/26 | 8/18 | 6/19 | 16/21 | 13/24 |
| SWM | 20/23 | 13/20 | 10/21 | 6/20 | 19/27 | 8/17 |
| SIM | 9/21 | 22/26 | 3/17 | 11/20 | 13/25 | 8/19 |
| Season | Norton Bridge | | | Manampitiya | | |
| | Dry | Normal | Wet | Dry | Normal | Wet |
| NEM | 5/20 | 25/29 | 2/15 | 22/23 | 11/25 | 1/16 |
| FIM | 3/20 | 14/23 | 14/20 | 9/21 | 20/24 | 5/19 |
| SWM | 16/22 | 9/22 | 9/20 | 2/21 | 26/27 | 6/16 |
| SIM | 7/22 | 15/22 | 11/20 | 17/25 | 13/20 | 7/19 |

**Table A. 5** Random forest model results. Highlighted cells indicate where there may be information content with
respect to forecasting either dry or wet anomaly classes

| Season | Morape | | | Peradeniya | | |
|--------|--------|--------|--------|--------|--------|--------|
| | Dry | Normal | Wet | Dry | Normal | Wet |
| NEM | 12/21 | 12/29 | 5/14 | 9/20 | 17/31 | 5/13 |
| FIM | 8/19 | 14/25 | 10/20 | 7/20 | 17/28 | 6/16 |
| SWM | 11/24 | 6/21 | 11/19 | 11/23 | 1/19 | 13/22 |
| SIM | 8/19 | 16/28 | 2/17 | 5/25 | 9/19 | 6/20 |
| Season | Randenigala | | | Bowatenna | | |
| | Dry | Normal | Wet | Dry | Normal | Wet |
| NEM | 10/24 | 8/25 | 4/15 | 16/24 | 6/19 | 11/21 |
| FIM | 9/20 | 8/25 | 8/19 | 16/21 | 14/25 | 4/18 |
| SWM | 9/21 | 14/24 | 6/19 | 14/25 | 7/21 | 5/18 |
| SIM | 15/24 | 6/21 | 7/19 | 3/21 | 14/26 | 11/17 |
| Season | Laxapana | | | Norwood | | |
| | Dry | Normal | Wet | Dry | Normal | Wet |
| NEM | 3/19 | 11/24 | 9/21 | 9/19 | 16/28 | 8/17 |
| FIM | 1/20 | 18/26 | 1/18 | 8/19 | 10/21 | 12/24 |
| SWM | 19/23 | 9/20 | 4/21 | 6/20 | 15/27 | 4/17 |
| SIM | 10/21 | 12/26 | 3/17 | 8/20 | 14/25 | 8/19 |
| Season | Norton Bridge | | | Manampitiya | | |
| | Dry | Normal | Wet | Dry | Normal | Wet |
| NEM | 11/20 | 12/29 | 6/15 | 14/23 | 10/25 | 5/16 |
| FIM | 7/21 | 8/23 | 8/20 | 10/21 | 11/24 | 6/19 |
| SWM | 9/22 | 6/22 | 8/20 | 6/21 | 17/27 | 5/16 |
| SIM | 13/22 | 9/22 | 9/20 | 15/25 | 8/20 | 7/19 |

[Figure]

**Figure A 8** Identifying relationships between two rainfall classes (dry, not dry) and MEI and DMI values using classification tree models for wet zone sub basins for SWM and SIM seasons. (a) Morape (b) Peradeniya (c) Laxapana (d) Norwood (e) Norton Bridge

[Figure]

**Figure A 9** Identifying relationships between two rainfall classes (dry, not dry) and MEI and DMI values using classification tree models for dry and intermediate zone sub basins for NEM and SIM seasons. (f) Randenigala (g) Bowatenna (h) Manampitiya

**Table A. 6** Classification tree model results for major rainfall season to the sub basins.

| Season | Morape | | Peradeniya | | Laxapana | | Norwood | | Norton Bridge | |
|---|---|---|---|---|---|---|---|---|---|---|
| | Dry | Not dry | Dry | Not dry | Dry | Not dry | Dry | Not dry | Dry | Not dry |
| SWM | 21/24 | 22/40 | 18/23 | 26/41 | 19/23 | 27/41 | 12/20 | 34/44 | 19/22 | 29/42 |
| SIM | 10/19 | 39/45 | 12/19 | 30/45 | 8/21 | 36/43 | 11/20 | 38/44 | 13/22 | 36/42 |

| Season | Randenigala | | Bowatenna | | Manampitiya | |
|---|---|---|---|---|---|---|
| | Dry | Not dry | Dry | Not dry | Dry | Not dry |
| NEM | 11/24 | 31/40 | 14/24 | 34/40 | 13/23 | 34/41 |
| SIM | 23/24 | 22/40 | 15/21 | 32/43 | 22/25 | 26/39 |

**Table A. 7** Random forest model results.

| Season | Morape | | | Peradeniya | | |
|---|---|---|---|---|---|---|
| | Dry | Not dry | OOB Error | Dry | Not dry | OOB Error |
| NEM | 10/21 | 33/43 | 33% | 8/20 | 34/44 | 34% |
| FIM | 5/19 | 36/45 | 36% | 6/20 | 37/44 | 33% |
| SWM | 11/24 | 29/40 | 38% | 11/23 | 28/41 | 39% |
| SIM | 5/19 | 39/45 | 33% | 5/19 | 37/45 | 34% |
| Season | Randenigala | | | Bowatenna | | |
| | Dry | Not dry | OOB Error | Dry | Not dry | OOB Error |
| NEM | 8/24 | 31/40 | 39% | 15/24 | 33/40 | 25% |
| FIM | 6/20 | 39/44 | 30% | 13/21 | 38/43 | 20% |
| SWM | 7/21 | 38/43 | 30% | 11/25 | 29/39 | 38% |
| SIM | 13/24 | 31/40 | 31% | 6/21 | 35/43 | 36% |
| Season | Laxapana | | | Norwood | | |
| | Dry | Not dry | OOB Error | Dry | Not dry | OOB Error |
| NEM | 8/20 | 37/45 | 30% | 10/19 | 39/45 | 23% |
| FIM | 7/20 | 37/44 | 31% | 8/19 | 39/45 | 26% |
| SWM | 12/23 | 27/41 | 39% | 7/20 | 37/44 | 31% |
| SIM | 9/21 | 34/43 | 33% | 7/20 | 37/44 | 31% |
| Season | Norton Bridge | | | Manampitiya | | |
| | Dry | Not dry | OOB Error | Dry | Not dry | OOB Error |
| NEM | 9/20 | 36/44 | 30% | 13/23 | 33/41 | 28% |
| FIM | 5/21 | 35/43 | 38% | 8/21 | 35/43 | 33% |
| SWM | 9/22 | 32/42 | 36% | 5/16 | 34/43 | 39% |
| SIM | 10/22 | 36/42 | 28% | 16/25 | 34/39 | 22% |

[Figure]

**Figure A 10** Random forest importance of variable to identify the dry and not dry classes of rainfall anomalies